# Characterization of Gaussian Universality Breakdown in High-Dimensional Empirical Risk Minimization

Mohamed Chiheb Yaakoubi [1]   Cosme Louart [1]   Malik Tiomoko [2]   Zhenyu Liao [3]

## Abstract

We study high-dimensional convex empirical risk minimization (ERM) under general non-Gaussian data designs. By heuristically extending the Convex Gaussian Min–Max Theorem (CGMT) to non-Gaussian settings, we derive an asymptotic min–max characterization of key statistics, enabling approximation of the mean $\mu_{\hat{\theta}}$ and covariance $C_{\hat{\theta}}$ of the ERM estimator $\hat{\theta}$. Specifically, under a concentration assumption on the data matrix and standard regularity conditions on the loss and regularizer, we show that for a test covariate $x$ independent of the training data, the projection $\hat{\theta}^{\top} x$ approximately follows the convolution of the (generally non-Gaussian) distribution of $\mu_{\hat{\theta}}^{\top} x$ with an independent centered Gaussian variable of variance $\mathrm{tr}\big(C_{\hat{\theta}} \, \mathbb{E}[xx^{\top}]\big)$. This result clarifies the scope and limits of Gaussian universality for ERMs. Additionally, we prove that any $\mathcal{C}^2$ regularizer is asymptotically equivalent to a quadratic form determined solely by its Hessian at zero and gradient at $\mu_{\hat{\theta}}$. Numerical simulations across diverse losses and models are provided to validate our theoretical predictions and qualitative insights.

## 1. Introduction

Given a training set of $n$ samples $(x_1, y_1), \ldots, (x_n, y_n) \in \mathbb{R}^p \times \mathcal{Y}$, we consider convex regularized empirical risk minimization (ERM) problems of the form

$$\hat{\theta} = \arg\min_{\theta \in \mathbb{R}^p} \left\{ \frac{1}{n} \sum_{i=1}^{n} \mathcal{L}_{y_i}(x_i^{\top}\theta) + \rho(\theta) \right\}, \qquad (1)$$

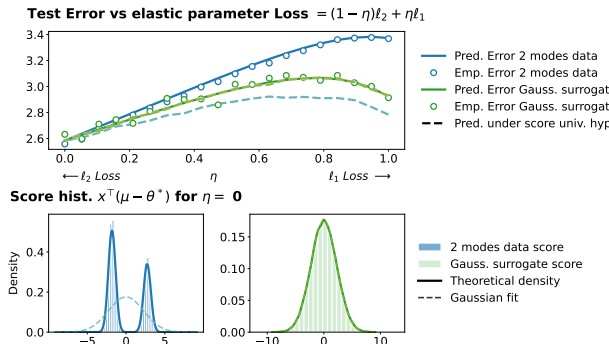

Figure 1. **Breakdown of performance and score universality through Elastic loss.** *Top:* Empirical generalization error and theoretical prediction of a regression task for a teacher–student model $y = \theta^{*\top} x + \varepsilon$, where $x$ follows a bimodal distribution ($p = 30$, $n = 50$). Performance universality is tested through comparison with Gaussian surrogate data that share the same mean and covariance as $x$. *Bottom:* Score histograms for $\mathcal{L}_y(\hat{\theta}^{\top} x) = \|\hat{\theta}^{\top} x - y\|^2$ at $\eta = 0$, along with their theoretical prediction and the Gaussian fit used to test the score universality assumption in the top panel. Performance and score universality are only satisfied when $\eta = 0$, i.e. for the $\ell_2$ loss. Note that the scores do not have to be Gaussian for performance universality to hold, as is the case for the $\ell_2$ loss.

where $\rho \colon \mathbb{R}^p \to \mathbb{R}$ is a convex regularizer and, for each $y \in \mathcal{Y}$, $\mathcal{L}_y \colon \mathbb{R} \to \mathbb{R}$ is a convex loss function. We focus on the high-dimensional regime in which both the sample size $n$ and the dimension $p$ are large, with typically $p = \Theta(n)$. As the minimizer of a random objective in (1), the solution $\hat{\theta}$ is itself random, through its dependence on the training set.

A central object for prediction is the *test score* $x^{\top}\hat{\theta}$, evaluated at a fresh covariate vector $x$ independent of the training data. Many performance metrics for regression and classification (e.g., prediction or classification error) can be expressed as functions of (the distribution of) $x^{\top}\hat{\theta}$. The goal of the present paper is to develop a general framework to characterize its asymptotic law, and hence to predict the performance of ERM, in high dimensions, under *general* (possibly *non-Gaussian*) data and convex losses.

Several complementary theoretical toolkits have been developed to yield sharp high-dimensional predictions for

[1]School of Data Science, The Chinese University of Hong Kong, Shenzhen, China [2]Huawei Noah's Ark Lab, Huawei Technologies, Paris, France [3]School of Electronic Information and Communications, Huazhong University of Science & Technology, China. Correspondence to: Cosme Louart <cosmelouart@cuhk.edu.cn>.

*Proceedings of the 43$^{rd}$ International Conference on Machine Learning*, Seoul, South Korea. PMLR 306, 2026. Copyright 2026 by the author(s).

ERM under various random designs. Random matrix theory (RMT) provides a natural framework when $p = \Theta(n)$. Starting with the seminal work of Karoui et al. (2013), RMT yields precise deterministic equivalents for statistics of $\hat{\theta}$:

$$\mu_{\hat{\theta}} := \mathbb{E}[\hat{\theta}] \quad \text{and} \quad C_{\hat{\theta}} := \mathbb{E}[\hat{\theta}\hat{\theta}^\top] - \mu_{\hat{\theta}}\mu_{\hat{\theta}}^\top. \quad (2)$$

in specific models, often through implicit fixed-point characterizations (Bean et al., 2013; Karoui, 2018; Couillet et al., 2018; Louart et al., 2018; Seddik et al., 2020; Couillet & Liao, 2022; Dobriban & Wager, 2018). Approximate message passing (AMP) and its associated state evolution offer another route to exact asymptotic formulas for the risk and distribution of high-dimensional $M$-estimators (Bayati & Montanari, 2011; Rangan, 2011; Donoho & Montanari, 2016; Loureiro et al., 2021b; Han & Shen, 2022; Dandi et al., 2023; Montanari & Saeed, 2022; Pesce et al., 2023). Finally, the convex Gaussian min–max theorem (CGMT) reduces certain high-dimensional convex optimization problems with Gaussian design matrices to lower-dimensional auxiliary optimization (AO) problems, whose optimal values (and in many cases optimizers) closely track those of the original program (Gordon, 1988; Thrampoulidis et al., 2014; 2015; 2018; Loureiro et al., 2021a; Akhtiamov et al., 2024; Mallory et al., 2025; Bosch & Panahi, 2025).

Although developed from different starting points, these approaches ultimately lead to essentially the *same* system of fixed-point equations under Gaussian design. Moreover, the fixed-point equations arising in RMT and AMP can often be interpreted as reformulations of the KKT conditions of the CGMT min–max/AO problem—a viewpoint that will also play a role in this paper. What differs across methods is therefore primarily the class of models under which the corresponding arguments can be rigorously justified.

Despite these differences, all three approaches typically encounter a recurring obstacle. Either the analysis is restricted to the special case of Gaussian design, or, when extending to non-Gaussian settings, one ultimately relies on some form of *Gaussian universality* argument. Such arguments seek to "replace" the data *or* the test score by a Gaussian proxy with matching first-and second-order moments. In this paper, we provide a novel analysis framework that makes this universality step *explicit*, and allows us to precisely characterize when it holds and when it breaks down. As shown in Figure 1, where we experiment with a mixture model using bimodal features, Gaussian universality does not hold in all cases. The left panel shows that the score distribution is non-Gaussian, which leads to a mismatch with theoretical predictions. In the right panel, we demonstrate that replacing the data with a Gaussian distribution with matching moments causes a significant divergence between the theoretical and empirical classification error. In Appendix G, we provide a similar analysis on real-world MNIST data with further explanations.

## 1.1. Main Contributions

Our starting point is an extension of the CGMT to a broad class of non-Gaussian designs to yield a set of powerful consequences that recover and generalize a large fraction of the existing exact-asymptotics literature for smooth losses. Our main contributions are as follows:

1. **(Theorem 3.3)** We show, under mild technical conditions, that for a given data model $(x, y)$ and loss $\mathcal{L}_y$, any $\mathcal{C}^2$ regularizer $\rho : \mathbb{R}^p \to \mathbb{R}$ is asymptotically equivalent, for the statistical behavior of the ERM solution $\hat{\theta}$ in (1), to the following *explicit quadratic surrogate*:

$$\rho_q(\theta) := \nabla\rho(\mu_{\hat{\theta}})^\top \theta + \frac{1}{2}(\theta - \mu_{\hat{\theta}})^\top H_\rho \theta, \quad (3)$$

where $\nabla\rho$ is the gradient of $\rho$, $H_\rho := \nabla^2\rho(0)$, the Hessian of $\rho$ evaluated at $\theta = 0$, and $\mu_{\hat{\theta}}$ in (2).

2. **(Theorem 4.3)** In a spirit similar to CGMT, we characterize, for possibly non-Gaussian data, the asymptotic behavior of the ERM solution $\hat{\theta}$ in (1), via the deterministic equivalent objective $\mathcal{J}$, defined, for any $\mu \in \mathbb{R}^p$ and $\alpha, \nu, \beta, \kappa > 0$ as:

$$\mathcal{J}(\mu, \alpha, \kappa; \beta, \nu) \quad (4)$$
$$= \frac{\beta^2\kappa}{2} + \mathbb{E}\big[e_{\mathcal{L}_y}\big(\mu^\top x + \alpha z; \kappa\big)\big] + \rho(\mu)$$
$$\quad - \frac{\nu\alpha^2}{2} - \frac{\beta^2}{2n}\text{tr}\Big(C_x(\nu C_x + H_\rho)^{-1}\Big),$$

where $C_x$ is the covariance of $x$, $z \sim \mathcal{N}(0, 1)$ is independent of $(x, y)$, and $e_{\mathcal{L}_y} : \mathbb{R} \times \mathbb{R} \to \mathbb{R}$ denotes the Moreau envelope defined, for any $u \in \mathbb{R}, \kappa > 0$ as:

$$e_{\mathcal{L}_y}(u; \kappa) := \min_{v \in \mathbb{R}}\left\{\frac{1}{2\kappa}(u - v)^2 + \mathcal{L}_y(v)\right\}. \quad (5)$$

In particular, define $\mu_* \in \mathbb{R}^p, \alpha_* \in \mathbb{R}$ solution to the following min–max problem:

$$(\mu_*, \alpha_*) := \underset{\mu \in \mathbb{R}^p, \ \alpha > 0}{\arg\min} \min_{\kappa > 0} \max_{\beta, \nu > 0} \mathcal{J}(\mu, \alpha, \kappa, \beta, \nu), \quad (6)$$

we show that $(\mu_*, \alpha_*)$ provides sharp approximations to $(\mu_{\hat{\theta}}, \sqrt{\text{tr}(C_{\hat{\theta}}C_x)})$, for $C_{\hat{\theta}}$ the covariance of $\hat{\theta}$ in (2), and are sufficient to characterize the ERM performance.

3. **(Theorem 6.1)** We show that asymptotically the test score approximates

$$x^\top\hat{\theta} \approx x^\top\mu_* + \alpha_* z,$$

with $z \sim \mathcal{N}(0, 1)$ independent of $x$. Crucially, this implies that the test score has an asymptotically Gaussian distribution *if and only if* the one-dimensional projection $x^\top\mu_*$ is Gaussian.

4. **(Theorem 7.1)** We show that when the dependence of $x$ on $y$ is confined to a subspace $F \subset \mathbb{R}^p$, then $\mu_{\hat{\theta}} \in F + \mathbb{R}(\nabla\rho(\mu_*) - H_\rho\mu_*)$. This result applies both to regression and classification settings.

## 1.2. Related Work on Gaussian Universality

Since the literature uses the term "universality" in several inequivalent ways, we explicitly distinguish three distinct types of *Gaussian universality*:

1. *performance universality* holds if key performance metrics such as prediction risk or classification error remain asymptotically the same as if the design vectors $(x_1, \ldots, x_n)$ were replaced by Gaussian covariates with matching mean and covariance.

2. *score universality* holds if, for an independent test vector $x$, the scalar score $x^\top\hat{\theta}$ converges in distribution to a Gaussian random variable; and

3. *minimizer universality* holds if the ERM solution $\hat{\theta}$ is asymptotically Gaussian.

Over the past decade, these questions have been investigated across a variety of settings; we summarize the resulting landscape in Table 1. For completeness, we additionally include two supplementary categories, distinguishing (i) works that exhibits explicit breakdowns of universality, and (ii) works that derive performance predictions based on first-order statistics. Indeed, a number of previous efforts (Korada & Montanari, 2011; Panahi & Hassibi, 2017; Han & Shen, 2022; Montanari & Saeed, 2022; Dandi et al., 2023; Mallory et al., 2025) establish performance universality by leveraging Gaussian design surrogates to characterize asymptotic behavior under non-Gaussian data. These results typically aim to establish equivalence Gaussian and non-Gaussian models, rather than to provide explicit deterministic formulas for performance in the high-dimensional regime $p = \Theta(n)$—which is the specific focus of the bottom row of Table 1.

Among the three notions, *performance universality* has the most direct practical implications and has been studied extensively, starting with the use of Lindeberg method (Chatterjee, 2006) in Korada & Montanari (2011); Panahi & Hassibi (2017); Han & Shen (2022); Goldt et al. (2022). Three primary categories of assumptions/settings are known to yield performance universality:

1. **Gaussian covariates**: when $x$ is Gaussian, precise characterizations are available (Karoui et al., 2013; Bean et al., 2013; Karoui, 2018; Couillet et al., 2018; Mai & Liao, 2020). Note, however, that Gaussian *mixtures* may invalidate these conclusions, see Pesce et al. (2023).

2. **Squared $\ell_2$ loss**: in the special case of squared loss, many *performance universality* results have been obtained (Korada & Montanari, 2011; Panahi & Hassibi, 2017; Han & Shen, 2022). In Section 5, we demonstrate that for squared loss, performance universality can hold *even when score universality fails*.

3. **Asymptotic score Gaussianity**: Recent works (Montanari & Saeed, 2022; Mallory et al., 2025) derive performance universality by assuming *score universality*, a condition we discuss below in greater detail.

To establish performance universality, Montanari & Saeed (2022) and subsequently Mallory et al. (2025) rely on assumptions that, while seemingly mild, effectively enforce a central limit theorem for $x^\top\hat{\theta}$ (identified in Table 1 as *score universality*). Specifically, they assume the existence of a ("unavoidable") subset $S \subset \mathbb{S}^{p-1}$ such that for any bounded Lipschitz function $\phi : \mathbb{R} \to \mathbb{R}$ and any Gaussian vector $g$ matching the first two moments of $x$,

$$\sup_{u \in S} \left| \mathbb{E}[\phi(u^\top x)] - \mathbb{E}[\phi(u^\top g)] \right| \xrightarrow[n \to \infty]{} 0.$$

If one can further guarantee that $\hat{\theta}/\|\hat{\theta}\| \in S$ with high probability, it follows that

$$\mathbb{E}[\phi(\hat{\theta}^\top x)] - \mathbb{E}[\phi(\hat{\theta}^\top g)] \xrightarrow[n \to \infty]{} 0. \tag{7}$$

However, this condition constitutes a significant limitation. It has been observed in several settings (Hu & Lu, 2022; Pesce et al., 2023; Mai & Liao, 2025) that the direction of the estimator may align with specific structural features of the data, resulting in $\mu_{\hat{\theta}}/\|\mu_{\hat{\theta}}\| \notin S$. In such cases, score universality—and consequently the universality of performance metrics derived via this route—breaks down. A central contribution of this paper is to characterize, as broadly as possible, the conditions under which such *Gaussian universality breakdown* occurs (Figure 1 and Figure 2 in the appendix show two examples where both score and performance universality are invalidated).

Finally, *minimizer universality* has received comparatively less attention, despite being plausibly generic for smooth $\mathcal{L}_{y_i}$ and $\rho$. The first-order optimality condition for (1) is

$$0 = \frac{1}{n}\sum_{i=1}^{n} \mathcal{L}'_y(x_i^\top\hat{\theta})\, x_i + \nabla\rho(\hat{\theta}).$$

Using the quadratic approximation of $\rho$ from (3), this yields:

$$H_\rho(\hat{\theta} - \mu_{\hat{\theta}}) \approx -\nabla\rho(\mu_{\hat{\theta}}) - \frac{1}{n}\sum_{i=1}^{n} \mathcal{L}'_{y_i}(x_i^\top\hat{\theta})\, x_i. \tag{8}$$

This suggests that $\hat{\theta}$ behaves as an average of many (weakly dependent) random vectors, and may therefore satisfy a

*Table 1.* **Comparison of related works and our method. Color code:** **Light Red** = RMT, **Light Green** = CGMT, **Light Blue** = AMP, **Light Purple** = Lindeberg-type arguments (and variants). **Notation:** ✓ indicates that the corresponding work covers the feature/setting; **R** denotes a proved statement; **A** denotes an assumption. Parenthesized **(R)** indicates that the statement holds only under a subset of the assumptions listed in that column. Note that **R** entries may rely on different baseline assumptions across columns and are therefore not always directly comparable. Papers are grouped within a single column when they share a closely related framework and essentially the same conclusions, although some modeling assumptions may differ.

| | (KOR11) | (ELK13) (BEA13) | (STO13) (THR18) | (DON16) | (PAN17) (HAN23) | (DOB18) | (ELK18) | (COU18) (MAI20) | (LOU21a) (LOU21b) | (MON22) (DAN23) | (HU22) | (PES23) | (ADO24) | (MAI25) | (AKH24) (BOS25) | (MAL25) | Our Method |
|---|---|---|---|---|---|---|---|---|---|---|---|---|---|---|---|---|---|
| **Statistics of $x$** | | | | | | | | | | | | | | | | | |
| $\mu_x \neq 0$ (non-zero mean) | | | | | | ✓ | | ✓ | ✓ | ✓ | | ✓ | ✓ | ✓ | ✓ | | ✓ |
| $C_x \neq \sigma^2 I_p$ (correlated features) | | | | | | ✓ | | ✓ | ✓ | ✓ | | ✓ | | ✓ | ✓ | ✓ | ✓ |
| Multiple statistics | | | | | | | | ✓ | ✓ | ✓ | | ✓ | | ✓ | ✓ | | ✓ |
| **Law type, default:** $x \sim \mathcal{N}(\mu_x, C_x)$ | | | | | | | | | | | | | | | | | |
| iid non-Gaussian entries | ✓ | | | | ✓ | ✓ | ✓ | | | ✓ | ✓ | | | | | ✓ | ✓ |
| Elliptical / linear factor | | | | | | | ✓ | | | ✓ | ✓ | | | ✓ | ✓ | | ✓ |
| Random features | | | | | | | | | | ✓ | ✓ | | | | | ✓ | ✓ |
| Heavy-tailed component | | | ✓ | | ✓ | | | | | | | | ✓ | | | | |
| **Dependence** $x \leftrightarrow y$ | | | | | | | | | | | | | | | | | |
| $y = \theta^{\star\top} x + \varepsilon$ | ✓ | ✓ | ✓ | ✓ | ✓ | ✓ | ✓ | | | ✓ | ✓ | ✓ | ✓ | | ✓ | ✓ | ✓ |
| $y$ is class label of $x$ | | | | | ✓ | | | ✓ | ✓ | ✓ | | | | ✓ | ✓ | | ✓ |
| $y = f(\theta^{\star\top} x)$ | | | | | | | | | | ✓ | ✓ | ✓ | | | | ✓ | ✓ |
| **Loss (default:** $\ell_2$**)** | | | | | | | | | | | | | | | | | |
| General convex on score | | ✓ | ✓ | ✓ | | | ✓ | ✓ | ✓ | ✓ | ✓ | ✓ | ✓ | ✓ | ✓ | | ✓ |
| **Regularization, default: None** | | | | | | | | | | | | | | | | | |
| Ridge ($\ell_2$) | | | ✓ | | ✓ | ✓ | ✓ | ✓ | ✓ | ✓ | ✓ | ✓ | ✓ | ✓ | ✓ | ✓ | ✓ |
| Lasso ($\ell_1$) | ✓ | | ✓ | | ✓ | | | | ✓ | | | ✓ | | | ✓ | | |
| General convex | | | ✓ | | ✓ | | | | ✓ | ✓ | ✓ | ✓ | ✓ | | ✓ | | ✓ |
| **Minimizer universality** | | R | | R | | | R | | | | | | | | | | **A** |
| **Score universality** | | R | | | | | | | | A | | | | | | A | **(R)** |
| **Performance universality** | R | R | | R | R | R | R | R | | R | (R) | (R) | (R) | (R) | | R | **(R)** |
| Non-universality **breakdown** | | | | | | | | | | | R | R | | R | | | **R** |
| Deterministic perf. **prediction** | | R | R | R | | R | R | R | R | | R | R | R | R | R | | **R** |

*Assumptions* (row group label) — *Results* (row group label)

Lindeberg-type central limit theorem. Motivated by this heuristic, we introduce minimizer universality as a standing assumption (Assumption 4) to establish full characterization of the test score. Moreover, it must be noted that this assumption is *not* required for the *non-Gaussian* CGMT-based pipeline in our Theorem 4.3.

### 1.3. Notations and Organization of the Paper

Throughout the paper, we adopt the notations introduced in (1). We denote the data matrices by $X = (x_1, \ldots, x_n) \in \mathbb{R}^{p \times n}$ and $Y = (y_1, \ldots, y_n) \in \mathcal{Y}^n$. A test sample independent of the training data $(X, Y)$ (and thus of $\hat{\theta}$) is generically denoted by $(x, y)$. Many theoretical formulas involve a standard Gaussian random variable independent of all data, which we denote by $z \sim \mathcal{N}(0, 1)$.

The Euclidean norm of a vector in $\mathbb{R}^p$ is denoted by $\|\cdot\|$. For matrices and tensors, $\|\cdot\|$ denotes the $\ell_2$ operator norm (spectral norm); specifically, for a tensor $T \in \mathbb{R}^{d_1 \times \cdots \times d_k}$, we define $\|T\| := \sup_{\|h_i\|=1} |T[(h_i)_{i \in [k]}]|$. For a matrix $M \in \mathbb{R}^{p \times n}$, we denote the Frobenius norm by $\|M\|_F = \sqrt{\operatorname{tr}(M M^\top)}$ and the nuclear norm by $\|M\|_* = \operatorname{tr}((M M^\top)^{\frac{1}{2}})$.

The remainder of this paper is organized as follows. Section 2 introduces the concentration and regularity assumptions used throughout the paper. Section 3 shows that, under suitable smoothness assumptions, the regularizer can be asymptotically replaced by a quadratic surrogate depending only on its local second-order structure. Section 4 presents a conditional extension of the CGMT framework to concentrated non-Gaussian designs and derives the corresponding deterministic min–max and fixed-point characterizations for the asymptotic statistics of the ERM estimator. Section 5 illustrates the framework on ridge regression under a linear model, in which case performance universality can hold even when score universality fails. Section 6 studies the asymptotic law of the test score and characterizes precisely when Gaussian score universality holds or breaks down. Finally, Section 7 provides geometric confinement results for the asymptotic mean vector $\mu_*$, yielding interpretable sufficient conditions for score universality.

## 2. Preliminaries and Problem Setup

We assume that the training set consists of $n$ independent samples $(x_1, y_1), \ldots, (x_n, y_n)$ drawn from a common dis-

tribution on $\mathbb{R}^p \times \mathcal{Y}$.[1] To obtain finite-$n$ bounds, we adopt assumptions inspired by concentration-of-measure theory. As shown in (Seddik et al., 2020), such assumptions encompass a broad class of realistic designs, including those generated via Lipschitz transformations of Gaussian latent variables, as commonly encountered in generative models. We focus on the high-dimensional regime in which the dimension $p$ scales at most linearly with $n$ (i.e., $p = O(n)$). Throughout, we track the explicit dependence of our bounds on $n$, while absorbing quantities that do not affect this scaling into generic constants $C, c > 0$.

We describe concentration properties of $(X, Y)$ through the behavior of Lipschitz observables. To this end, we endow the product space $\mathbb{R}^{p \times n} \times \mathcal{Y}^n$ with the metric:

$$d\big((x_i, y_i)_{i \in [n]}, (x'_i, y'_i)_{i \in [n]}\big)^2 = \sum_{i=1}^n \|x_i - x'_i\|^2 + d_{\mathcal{Y}}(y_i, y'_i)^2,$$

where $d_{\mathcal{Y}}$ denotes a metric on $\mathcal{Y}$.[2]

**Assumption 1** (Concentration through Lipschitz observations). *There exist numerical constants $C, c > 0$, independent of $n$, such that $p \leq Cn$, $\|\mathbb{E}[x]\| \leq C$, and for every 1-Lipschitz function $f : (\mathbb{R}^p \times \mathcal{Y})^n \to \mathbb{R}$:*

$$\mathbb{P}\big(\big|f(X, Y) - \mathbb{E}[f(X, Y)]\big| \geq t\big) \leq Ce^{-ct^2}. \quad (9)$$

Assumption 1 holds for nonlinear regression model with data $x_i = \Phi(z_i)$ for $z_i \sim \mathcal{N}(0, I_p)$ and Lipschitz $\Phi$, and target $y_i = g(x_i, w_i)$ being a Lipschitz transformation ($g$) of $x_i$ and Gaussian noise $w_i$.[3] Similarly, a binary classification model satisfying Assumption 1 can be described by labels $y_i \in \{1, 2\}$ and class-conditional designs $x_i = \phi_{y_i}(z_i)$ where $\phi_1, \phi_2$ are Lipschitz maps.

We next impose regularity conditions on the loss and regularizer to ensure (i) the existence and uniqueness of the ERM solution $\hat{\theta}$ and (ii) that the concentration properties of the data in Assumption 1 transfers smoothly to $\hat{\theta}$.

**Assumption 2** (Regularity of loss and regularizer). *For all $y \in \mathcal{Y}$, the functions $\mathcal{L}_y : \mathbb{R} \to \mathbb{R}$ and $\rho : \mathbb{R}^p \to \mathbb{R}$ are convex and twice continuously differentiable. We assume that $\mathcal{L}_y(0)$ and $\|\nabla\rho(0)\|$ are bounded uniformly in $n$. Furthermore:*

1. *There exists $\lambda > 0$ such that $\nabla\rho$ is $\lambda$-Lipschitz, and for any $y \in \mathcal{Y}$, the derivative $\mathcal{L}'_y$ is $\lambda$-Lipschitz. additionally,*

*the map $y \mapsto \mathcal{L}'_y(u)$ is $\lambda$-Lipschitz (with respect to $d_{\mathcal{Y}}$) for any fixed $u$.*

2. *There exists $\kappa > 0$, independent of $p$ and $n$, such that $\nabla^2\rho(\theta) \succeq \kappa I_p$ for all $\theta \in \mathbb{R}^p$.*

Assumption 2 excludes non-smooth penalties such as the Lasso. While the Lasso can be analyzed via CGMT (Stojnic, 2013), the strict convexity and smoothness assumed here are required to ensure that $\hat{\theta}$ concentrates strongly and satisfies the weakly dependent average formulation (8).

These two assumptions suffice to establish the concentration of the minimizer $\hat{\theta}$, which is fundamental to our subsequent analysis of the score's asymptotic behavior.

**Theorem 2.1** (**Concentration of $\hat{\theta}$**). *Under Assumptions 1 and 2, there exist constants $C, c > 0$, independent of $n, p$, such that for any 1-Lipschitz $f : \mathbb{R}^p \to \mathbb{R}$,*

$$\mathbb{P}\big(\big|f(\hat{\theta}) - \mathbb{E}[f(\hat{\theta})]\big| \geq t\big) \leq Ce^{-cnt^2}.$$

Theorem 2.1, combined with the concentration property of $x$ (Assumption 1), provides bounds on the operator norm of the covariance matrices $C_x$ and $C_{\hat{\theta}}$ of $x$ and $\hat{\theta}$. This result will be further explored to establish the asymptotic behavior of test scores in Theorem 6.1.

**Corollary 2.2** (Louart (2024)). *Under Assumptions 1 and 2, we have $\|C_x\| = O(1)$ and $\|C_{\hat{\theta}}\| = O(1/n)$.*

## 3. Quadratic Universality for Smooth Regularizers

We now demonstrate that, on the scale relevant for high-dimensional fluctuations, the regularizer can be effectively approximated by a quadratic form derived from a second-order Taylor expansion around $\mu_{\hat{\theta}}$. To control the approximation error, we impose bounds on the operator norm of the third-derivative tensor $\nabla^3\rho$. When $\rho$ is separable, $\nabla^3\rho(\theta)$ is a diagonal tensor for all $\theta \in \mathbb{R}^p$, and it is natural to assume that the diagonal entries do not grow with the dimension $p$ (as is the case for the $\ell_2$ penalty). For non-separable regularizers, the Taylor approximation requires additional control over off-diagonal interactions.

**Assumption 3.** *The function $\rho : \mathbb{R}^p \to \mathbb{R}$ is $\mathcal{C}^3$ and there exists a constant $C > 0$ such that, for all $\theta \in \mathbb{R}^p$:*[4]

$$\|\nabla^3\rho(\theta)\| \leq C \quad and \quad \|\operatorname{OffDiag}(\nabla^3\rho(\theta))\| \leq Cp^{-1/2}.$$

Note that the condition on the off-diagonal tensor entries is weaker than the "weakly separable" hypothesis typically

---

[1]Our results naturally extend to multiclass settings; see Section A in the appendix.

[2]This includes, for instance, the discrete/Hamming metric commonly used for classification labels, as well as the absolute difference $|y - y'|$ when $\mathcal{Y} = \mathbb{R}$.

[3]In this setting, $(X, Y)$ is a Lipschitz transformation of the Gaussian vector $(z_{1:n}, w_{1:n})$. By standard Gaussian concentration results (Ledoux, 2005), Equation (9) holds.

[4]For a tensor $T$, $\|T\|$ denotes the operator norm $\sup_{\|u\|=1} |T[u, u, u]|$. For the off-diagonal part, we refer to the operator norm of the tensor with diagonal entries set to zero.

assumed in the literature; see, e.g., (Panahi & Hassibi, 2017). A second-order Taylor expansion of $\rho$ around $\mu_{\hat{\theta}}$ yields the following concentration bound for the remainder.

**Proposition 3.1.** *Under Assumptions 1, 2, and 3, there exist constants $C, c > 0$ such that for all $t \geq 0$:*

$$\mathbb{P}\Big(\Big|\rho(\theta) - \rho(\mu_{\hat{\theta}}) - \nabla\rho(\mu_{\hat{\theta}})^{\top}(\theta - \mu_{\hat{\theta}})$$
$$- \tfrac{1}{2}(\theta - \mu_{\hat{\theta}})^{\top}\nabla^2\rho(\mu_{\hat{\theta}})(\theta - \mu_{\hat{\theta}})\Big| \geq t\Big) \leq Ce^{-cnt^2}.$$

We saw in (4) that $\nabla^2\rho$ enters our asymptotic formulas primarily through the trace term:

$$\frac{1}{n}\mathrm{tr}(C_x Q_\mu), \quad \text{where} \quad Q_\mu := \big(\nabla^2\rho(\mu) + \nu C_x\big)^{-1}.$$

In this expression, one may replace $\nabla^2\rho(\mu)$ with its value at zero,

$$H_\rho := \nabla^2\rho(0),$$

without affecting the asymptotic limit, thanks to the following lemma.

**Lemma 3.2.** *Under Assumption 3, there exist constants $C, C' > 0$, independent of $n$ and $p$, such that for all $\theta \in \mathbb{R}^p$, $\|H_\rho - \nabla^2\rho(\theta)\|_F \leq C\|\theta\|$. Consequently:*

$$\frac{1}{n}\mathrm{tr}\big(C_x(Q_\mu - Q_0)\big) \leq \frac{C'}{\sqrt{n}}, \text{ with } Q_0 := \big(H_\rho + \nu C_x\big)^{-1}.$$

Combining these observations with Theorem 4.4 (which characterizes the asymptotic statistics of $\hat{\theta}$) leads to our universality result for regularization. Let $\hat{\theta}$ be the solution to Problem (1), and let $\hat{\theta}_q$ be the solution to the same problem where $\rho$ is replaced by the quadratic surrogate:

$$\rho_q(\theta) := a^{\top}\theta + \frac{1}{2}\theta^{\top}H_\rho\theta \quad \text{with } a := \nabla\rho(\mu_{\hat{\theta}}) - H_\rho\mu_{\hat{\theta}}.$$

chosen s.t. $\nabla\rho_q(\mu_{\hat{\theta}}) = \nabla\rho(\mu_{\hat{\theta}})$ and $\nabla^2\rho_q(\theta) = H_\rho$.

**Theorem 3.3 (Quadratic universality of regularization).** *Under Assumptions 1, 2 and 3:*

$$\Big|\mathbb{E}[x^{\top}\hat{\theta}] - \mathbb{E}[x^{\top}\hat{\theta}_q]\Big|, \Big|\mathbb{E}[(x^{\top}\hat{\theta})^2] - \mathbb{E}[(x^{\top}\hat{\theta}_q)^2]\Big| \xrightarrow[n\to\infty]{} 0.$$

This result is equivalent to Theorem 4.3 presented below.

# 4. Non-Gaussian CGMT and Implications for ERM Asymptotics

The Convex Gaussian Min–Max Theorem (CGMT) transforms a min–max problem involving a random Gaussian matrix $A \in \mathbb{R}^{p \times n}$ into an associated *auxiliary problem* whose randomness depends only on two Gaussian vectors $g \in \mathbb{R}^p$ and $h \in \mathbb{R}^n$ (Gordon, 1988). Originally used to

study the singular values of Gaussian matrices (Vershynin, 2012; Oymak & Tropp, 2018), this tool has become a cornerstone for analyzing high-dimensional risk minimization, beginning with (Stojnic, 2013; Thrampoulidis et al., 2014).

The first step is to rewrite the ERM problem (1) as a constrained optimization problem with an objective linear in the random design matrix $X$:

$$\min_{\theta \in \mathbb{R}^p, \, v \in \mathbb{R}^n} \left\{\frac{1}{n}\mathcal{L}_y(v) + \rho(\theta)\right\} \quad \text{s.t.} \quad v = X^{\top}\theta.$$

Introducing a dual variable $w \in \mathbb{R}^n$ for the constraint yields the Lagrangian:

$$\mathcal{L}(\theta, v, w) = w^{\top}(X^{\top}\theta - v) + \frac{1}{n}\mathcal{L}_y(v) + \rho(\theta).$$

Minimization over $v$ and maximization over $w$ commute, leading to the following min–max formulation suited for an application of the CGMT:

$$\hat{\theta} \in \arg\min_{\theta \in \mathbb{R}^p} \max_{w \in \mathbb{R}^n} \left\{\theta^{\top}Xw - \frac{1}{n}\mathcal{L}_y^*(nw) + \rho(\theta)\right\}, \quad (10)$$

where $\mathcal{L}_y^*$ denotes the Fenchel conjugate of $\mathcal{L}_y$, i.e., $\mathcal{L}_y^*(w) = \sup_{v \in \mathbb{R}^n}\{v^{\top}w - \mathcal{L}_y(v)\}$.

In the claim below, we isolate the specific properties of the generalized CGMT required for our derivations (analogous to Thrampoulidis et al. (2014, Theorem 4)).

**Claim 4.1 (Generalized CGMT for concentrated column designs).** *Let $C_1 \subset \mathbb{R}^p$ and $C_2 \subset \mathbb{R}^n$ be convex compact sets. For each $y \in \mathcal{Y}$, let $\psi_y : \mathbb{R}^p \times \mathbb{R}^n \to \mathbb{R}$ be a function that is convex in its first argument and concave in its second. For a matrix $A \in \mathbb{R}^{p \times n}$, vectors $\mu, a \in \mathbb{R}^p$, and $h \in \mathbb{R}^n$, we define the primal and auxiliary value functions:*

$$\Phi(A) := \min_{\theta \in C_1} \mathcal{V}_\Phi(\theta, A), \qquad \phi(a, h) := \min_{\theta \in C_1} \mathcal{V}_\phi(\theta, a, h),$$

*where the objectives are given by:*

$$\mathcal{V}_\Phi(\theta, A) := \max_{w \in C_2} \left\{\theta^{\top}Aw + \mu^{\top}Aw + \psi_y(\theta, w)\right\},$$

$$\mathcal{V}_\phi(\theta, a, h) := \max_{w \in C_2} \Big\{w^{\top}h\,\|C_x^{\frac{1}{2}}\theta\| + \|w\|\,\theta^{\top}a$$
$$+ \mu^{\top}a\,\|w\| + \psi_y(\theta, w)\Big\}.$$

*Let $\theta_\Phi(A)$ and $\theta_\phi(a, h)$ denote the minimizers of $\mathcal{V}_\Phi(\cdot, A)$ and $\mathcal{V}_\phi(\cdot, a, h)$, respectively.*

*With our notations and under Assumption 1, if we consider $g \sim N(0, I_n)$, independent with $(X, x)$ and assume:*

*1. $\forall \varepsilon > 0 : \mathbb{P}\big(\big|\phi(x, g) - \mathbb{E}[\phi(x, g)]\big| \geq \varepsilon\big) \xrightarrow[n\to\infty]{} 0$*

*2. $\forall \varepsilon > 0 : \mathbb{P}\big(\big|\|\theta_\phi(x, g)\| - \mathbb{E}\big[\|\theta_\phi(x, g)\|\big]\big| \geq \varepsilon\big) \xrightarrow[n\to\infty]{} 0$*

3. *There exists $\tau > 0$, independent of $p$ and $n$, such that for $n$ large enough, and $\theta \in \mathbb{R}^p$:*

$$\mathcal{V}_\phi(\theta; x, h) \geq \phi(x, h) + \tau\big(\|\theta\| - \|\theta_\phi(x, g)\|\big)^2, \quad a.s.;$$

*then, $\forall \varepsilon > 0$, $\mathbb{P}(|\Phi(X) - \phi(x, g)| \geq \varepsilon) \xrightarrow[n \to \infty]{} 0$ and:*

$$\mathbb{P}\big(\big|\|\theta_\Phi(X)\| - \mathbb{E}\big[\|\theta_\phi(x, g)\|\big]\big| \geq \varepsilon\big) \xrightarrow[n \to \infty]{} 0.$$

*Remark* 4.2. Recent CGMT results such as those in (Akhtiamov et al., 2024; Bosch & Panahi, 2025), while treating multiclass settings, do not explicitly model the dependence of $\psi$ on the label $y$, nor do they incorporate the deterministic shift parameter $\mu \in \mathbb{R}^p$. This distinction is not merely anecdotal; a central component of our approach is to leverage the concentration of $\hat{\theta}$ via the decomposition:

$$\hat{\theta} = \mu_{\hat{\theta}} + \theta_0, \tag{11}$$

and to analyze the minimization with respect to the fluctuation $\theta_0$ and the mean $\mu_{\hat{\theta}}$ separately. When $\mu_{\hat{\theta}}$ is fixed, we apply our claim with $\mu = C_x^{\frac{1}{2}}\mu_{\hat{\theta}}$, $\psi_y(\theta_0, w) = \frac{1}{n}\mathcal{L}_y^*(nw) + \rho(\mu_{\hat{\theta}} + C_x^{-\frac{1}{2}}\theta_0)$, and replace the random matrix $X$ with the whitened version $C_x^{-\frac{1}{2}}X$. This yields theoretical predictions for quantities such as $\|C_x^{1/2}\theta_0\|$, which in our setting converges to $\sqrt{\operatorname{tr}(C_x C_{\hat{\theta}})}$. In contrast, standard CGMT approaches would focus on estimating the total norm $\|C_x^{1/2}(\theta_0 + \mu_{\hat{\theta}})\|$, which is less amenable to the subsequent convex analysis machinery we employ.

One can verify that Claim 4.1 applies to (10) under decomposition (11), as conditions 1–3 are natural consequences of Assumptions 1 and 2 (with Theorem 2.1 directly supporting condition 2). The convex analysis derivation of Claim 4.1 (which is still only a conjecture), detailed in Appendix C, yields the result below.

**Theorem 4.3 (Min–max formulation of limiting asymptotics).** *Under Assumptions 1, 2, and 3:*

$$\|\mu_{\hat{\theta}} - \mu_*\| \xrightarrow[n \to \infty]{} 0 \quad and \quad \operatorname{tr}(C_x C_{\hat{\theta}}) - \alpha_*^2 \xrightarrow[n \to \infty]{} 0,$$

*where $(\mu_*, \alpha_*)$ are the saddle-point solutions to the min–max problem in (6).*

In plain words, Theorem 4.3 states that the high-dimensional ERM estimator can be "summarized" asymptotically by two deterministic quantities: $\mu_* \approx \mathbb{E}[\hat{\theta}]$ that captures the deterministic informative component of the estimator and $\alpha_*^2 \approx \operatorname{tr}(C_x C_{\hat{\theta}})$ that measures the scale of its random fluctuations. These quantities can be explicitly obtained from a deterministic saddle-point problem depending only on the data model $(x, y)$, the loss, and the local quadratic structure of the regularizer. In this sense, the random high-dimensional

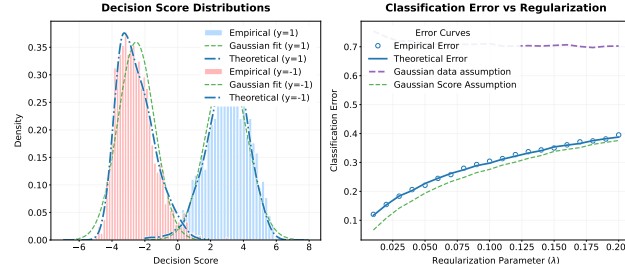

*Figure 2.* Universality breakdown on MNIST data. Setting described in Appendix G *Left:* Empirical histograms of decision scores for Class 0 (light blue) and Class 1 (light red) of $\hat{\theta}^\top x$, compared with a Gaussian approximation of matching mean and variance (green dashed) and with the corrected theoretical density(dashed blue). *Right:* Generalization performance. As expected, predictions based on Gaussian score universality (green dashed) fail to match empirical results.

optimization problem (1) admits a low-dimensional deterministic characterization.

A central consequence, developed further in Section 6, is that the test score $x^\top \hat{\theta}$ decomposes asymptotically into a structured component $x^\top \mu_*$ and an independent Gaussian fluctuation of variance $\alpha_*^2$. Gaussian universality therefore holds *if and only if* the informative projection $x^\top \mu_*$ itself is asymptotically Gaussian.

A key novelty relative to previous analyses is the explicit role of the vector $\mu_*$, which captures the (generally non-zero) asymptotic mean of $\hat{\theta}$ for general losses and data distributions. Theorem 4.3 reveals that the distribution of $x$ influences the asymptotics specifically through the Gaussian-convolved variable $\mu^\top x + \alpha z$ (see again Section 6).

For theoretical analysis and numerical computation, it is convenient to reformulate the min–max problem as a fixed-point system. For any $y \in \mathcal{Y}$, we define the mapping $\xi_y : \mathbb{R} \times \mathbb{R}_+ \to \mathbb{R}$ by:

$$\xi_y(u, \kappa) := u - \operatorname{prox}_{\kappa\mathcal{L}_y}(u), \tag{12}$$

where the proximal operator is given by $\operatorname{prox}_{\kappa\mathcal{L}_y}(u) := \arg\min_{w \in \mathbb{R}}\{\kappa\mathcal{L}_y(w) + \frac{1}{2}(u - w)^2\}$. We further define the resolvent $Q(\nu) := (\nu C_x + H_\rho)^{-1}$ and the functional $A(\nu) := \frac{1}{n}\operatorname{tr}(C_x Q(\nu) C_x Q(\nu))$.

**Theorem 4.4 (Fixed-point formulation).** *Under Assumptions 2 and 3, the parameters $(\mu_*, \alpha_*)$ defined in (6) satisfy the following system of equations:*

$$\kappa = \frac{1}{n}\operatorname{tr}\big(C_x Q(\nu)\big),$$

$$\nu = \frac{1}{\alpha_* \kappa}\mathbb{E}\big[z\,\xi_y\big(\mu_*^\top x + \alpha_* z, \kappa\big)\big],$$

$$\alpha_*^2 = \frac{A(\nu)}{\kappa^2} \mathbb{E}\Big[\xi_y\big(\mu_*^\top x + \alpha_* z, \kappa\big)^2\Big],$$

$$0 = \nabla\rho(\mu_*) + \frac{1}{\kappa}\mathbb{E}\big[x\,\xi_y\big(\mu_*^\top x + \alpha_* z, \kappa\big)\big].$$

*The expectations are taken with respect to $x, y$ and an independent $z \sim \mathcal{N}(0,1)$.*

In the specific case of squared loss with a linear model, Theorem 4.4 reduces to the two-equation system derived in Karoui et al. (2013); Bean et al. (2013); Donoho & Montanari (2016) (see Section 7). More broadly, it generalizes fixed-point characterizations previously established for regularized regression (Thrampoulidis et al., 2015) and high-dimensional classification in mixture models (Mai & Liao, 2020; 2025).

# 5. Performance Universality of Ridge Regression under Linear Model

In this section, we apply our framework to the specific case of squared loss with a linear data model:

$$\mathcal{L}_y(t) = \frac{1}{2}(t - y)^2, \qquad \text{where} \quad y = \theta^{*\top} x + \varepsilon. \quad (13)$$

Here, $\theta^* \in \mathbb{R}^p$ is a deterministic signal vector independent of $x$, and $\varepsilon \in \mathbb{R}$ is a random noise variable independent of $x$ satisfying:

$$\mathbb{E}[\varepsilon] = 0 \qquad \text{and} \qquad \sigma_\varepsilon^2 := \mathbb{E}[\varepsilon^2]. \quad (14)$$

This setting mirrors the one analyzed in Dobriban & Wager (2018), with the generalization that $X$ need not be a linear transformation of a random matrix with independent entries. Following the notation of the previous section, the proximal operator for the squared loss admits a closed-form expression, leading to:

$$\xi_y(u, \kappa) = \frac{\kappa}{1 + \kappa}(u - y).$$

This greatly simplifies the fixed-point equations in Theorem 4.4. Specifically, the relation between $\nu$ and $\kappa$ becomes explicit, yielding a single equation for $\nu_*$:

$$\nu_* = \frac{1}{1 + \kappa_*} = \frac{1}{1 + \frac{1}{n}\operatorname{tr}\big(C_x\, Q(\nu_*)\big)}. \quad (15)$$

Note that this identity is indeed satisfied for the $\ell_2$ norm on Figure 3. This is the classical fixed-point equation from Random Matrix Theory, known to possess a unique solution (Silverstein & Bai, 1995). Leveraging the quadratic universality established in Theorem 3.3, we adopt the quadratic formulation for $\rho$ below to derive a structural result.

**Corollary 5.1** (**Generalization error of Ridge regression**). *Consider the setting (13) with a quadratic regularizer*

**Description variables through elastic loss**

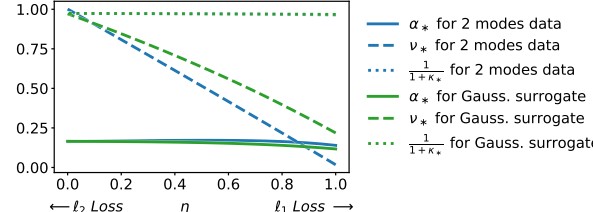

*Figure 3.* **Evolution of the description variables through elastic loss.** In the setting of Figure 1, we show the fixed-point solution variables $\alpha_*$, $\nu_*$ and $\frac{1}{1+\kappa_*}$. For the $\ell_2$ loss ($\eta = 0$), we have $\nu_* = \frac{1}{1+\kappa_*}$.

$\rho(\theta) = a^\top\theta + \frac{1}{2}\theta^\top H\theta$, *where $a \in \mathbb{R}^p$ and $H \in \mathbb{R}^{p\times p}$ is positive symmetric. Under Assumptions 1–3, let $\nu_*$ be the unique solution to Equation (15). Then:*

$$\mathbb{E}\Big[\mathcal{L}_y(x^\top\hat{\theta})\Big] - \frac{\nu_*^2 A(\nu_*)\sigma_\varepsilon^2 + \Delta(\nu_*)}{1 - \nu_*^2 A(\nu_*)} \xrightarrow[n\to\infty]{} 0,$$

*where, with notation $\Sigma_x := C_x + \mu_x\mu_x^\top$, we defined:*

$$\forall\nu \in \mathbb{R}: \qquad \mu(\nu) := Q(\nu)\left(\nu C_x\theta^* - a\right),$$
$$\Delta(\nu) := (\mu(\nu) - \theta^*)^\top\Sigma_x(\mu(\nu) - \theta^*).$$

Notably, the asymptotic generalization error depends on the distribution of $x$ solely through its first two moments ($C_x$ and $\mu_x$). This confirms that *performance universality* holds in this setting. However, note that there is no reason for the *score universality* to be satisfied, as $x^\top\hat{\theta}$ may be non-Gaussian depending on the distribution of $x^\top\theta^*$.

# 6. Score Universality Characterization

Unlike the score universality condition (7) assumed in Montanari & Saeed (2022) and Mallory et al. (2025), we are not aware of any empirical counterexamples to the following assumption in the proportional regime. See again discussion around (8) to understand the underlying heuristic. A proof of this result based on Chatterjee's normal-approximation method ((Chatterjee, 2008; Shao & Zhang, 2025)) is under preparation.

**Assumption 4** (Minimizer universality). *For any bounded measurable function $f : \mathbb{R} \to \mathbb{R}$ and any deterministic sequence $(u_n)_{n\in\mathbb{N}} \subset \mathbb{R}^p$ with $\|u_n\| = O(1)$:*

$$\mathbb{E}\big[f(u_n^\top\hat{\theta})\big] - \mathbb{E}\big[f(u_n^\top g)\big] \xrightarrow[n\to\infty]{} 0,$$

*where $g \sim \mathcal{N}(\mu_{\hat{\theta}}, C_{\hat{\theta}})$.*

Conditioning on an independent test covariate $x$, Assumption 4 implies the approximation:

$$x^\top\hat{\theta} \approx x^\top g \quad \Longrightarrow \quad x^\top g \mid x \sim \mathcal{N}\big(x^\top\mu_{\hat{\theta}}, x^\top C_{\hat{\theta}} x\big).$$

In fact, $x^\top A x$ concentrates with high probability around $\operatorname{tr}(\Sigma_x A)$ by Hanson–Wright inequality, see Adamczak (2015, Theorem 2.3), which leads to the following result.

**Theorem 6.1** (**Asymptotic behavior of test score**). *Under Assumptions 1–4, for any bounded measurable function $f : \mathbb{R} \times \mathcal{Y} \to \mathbb{R}$:*

$$\mathbb{E}\big[f(x^\top \hat{\theta}, y)\big] - \mathbb{E}\big[f(x^\top \mu_* + \alpha_* z, y)\big] \xrightarrow{n \to \infty} 0,$$

*where $(\mu_*, \alpha_*)$ are defined in Theorem 4.3 and $z \sim \mathcal{N}(0,1)$ is independent of $(x,y)$.*

Theorem 6.1 fully characterizes the score universality:

$$\text{``} x^\top \hat{\theta} \text{ Gaussian} \iff x^\top \mu_* \text{ Gaussian''}.$$

It is natural at this stage to relate our score characterization to the recent literature on Gaussian and conditional Gaussian equivalence for random feature models (Gerace et al., 2022; Dandi et al., 2023; Wen et al., 2025). A common formulation assumes the existence of a low-dimensional random vector $s \in \mathbb{R}^r$, with $r = O(1)$, such that

$$y = \eta(s, \varepsilon), \qquad x \mid s \sim \mathcal{N}\big(m(s), \Sigma(s)\big),$$

where $\varepsilon$ is independent of $(x, s)$. Under such a representation, our score formula yields

$$x^\top \hat{\theta} \approx m(s)^\top \mu_* + \sqrt{\mu_*^\top \Sigma(s) \mu_* + \alpha_*^2}\, z,$$

so that the fixed-point system of Theorem 4.4 reduces to expectations over the low-dimensional variable $s$. In particular, the non-Gaussian part of the score is entirely carried by the scalar quantity $m(s)^\top \mu_*$, while the remaining fluctuations are Gaussian. We leave a systematic study of this reduced system to future work, and instead focus below on a simpler structural consequence of our analysis, namely the confinement of $\mu_{\hat{\theta}}$ when only specific projections of $x$ are informative for predicting $y$.

## 7. Subspace Confinement of $\mu_{\hat{\theta}}$

As defined in Eq. (4), the vector $\mu_*$ is an implicit parameter that depends on the interplay between the alignment of the learning task and the distribution profile of $x$. This dependency makes the geometric interpretation of the score universality condition in Theorem 6.1 difficult. However, in structured signal-plus-noise models, the geometry of the problem imposes strong constraints on the location of $\mu_*$.

**Theorem 7.1.** *Consider the ERM problem (1) with a quadratic regularizer $\rho(\theta) = a^\top \theta + \theta^\top H \theta$, where $a \in \mathbb{R}^p$ and $H \in \mathbb{R}^{p \times p}$ is a positive symmetric matrix. Assume the data decomposes as $x = x_I + x_N$ almost surely, where $x_I$ lies in a subspace $F \subset \mathbb{R}^p$ and $x_N$ is a random vector independent of both $x_I$ and $y$. Then, the asymptotic mean vector $\mu_*$ defined in Theorem 4.3 satisfies:*

$$\mu_* \in F + \operatorname{span}(a).$$

This theorem confines the effective signal proxy $\mu_*$ to the subspace containing the informative part of the data and the regularization shift. This significantly simplifies the verification of score universality. Figure 4 below demonstrates the characterization of score universality provided in the following corollary.

**Corollary 7.2.** *In the setting of Theorem 7.1, assume further that for all $u \in F + \operatorname{span}(a)$, the projection $u^\top x$ follows a Gaussian distribution. Then, for any bounded measurable function $f : \mathbb{R} \times \mathcal{Y} \to \mathbb{R}$:*

$$\mathbb{E}\big[f(x^\top \hat{\theta}, y)\big] \xrightarrow{n \to \infty} \mathbb{E}\big[f(G, y)\big],$$

*where $G \sim \mathcal{N}\big(\mu_*^\top \mathbb{E}[x],\ \mu_*^\top C_x \mu_* + \alpha_*^2\big)$.*

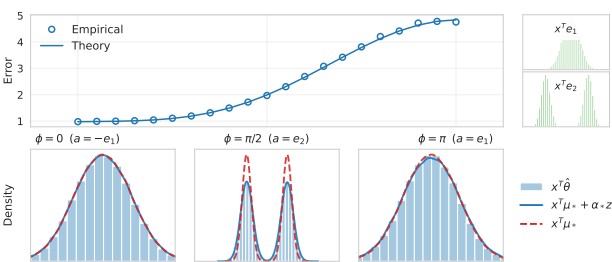

*Figure 4.* We examine different score distributions for various regularization functions $\rho : \theta \mapsto a^\top \theta + \|\theta\|^2$, where $a = (-\cos(\phi), \sin(\phi), 0, \ldots, 0)$ for some angle $\phi = 0, \frac{\pi}{2}, \pi$ (from *bottom left* to *bottom right*). We use the squared loss $\mathcal{L}_y(z) = (z - y)^2$, with $y = \theta^{*\top} x + \varepsilon$, where $\theta^* = e_1$. For all $i \in [p] \setminus 2$, $x^\top e_i \sim \mathcal{N}(0,1)$, while $x^\top e_2$ follows a bimodal distribution. According to Corollary 7.2, the score is Gaussian when $x^\top a$ is Gaussian, which occurs here only at the extremal points of the graph ($\phi \in \{0, \pi\}$). The error *(top left)* is minimized when $a = -e_1$ (i.e. $\phi = 0$), in which case $\theta^*$ minimizes $\rho$.

## Software and Data

The code to reproduce our experiments is available at: https://github.com/cosmital/Empirical-risk-minimization-asymptotics.

## Acknowledgements

This work is supported by the National Natural Science Foundation of China (Research fund for international young scientists grant W2533014), Huawei Strategic Research Institute, the Shenzhen Municipal Government Talent Research Start-up Funding, the National Key Research and Development Program of China (No. 2025YFA1018600), the National Natural Science Foundation of China (via fund NSFC-12571561), and the Fundamental Research Support Program of HUST (2025BRSXB0004).

## Impact Statement

This paper presents work whose goal is to advance the field of Machine Learning. There are many potential societal consequences of our work, none of which we feel must be specifically highlighted here.

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

## A. Multiclass ERM asymptotics

Classification settings are simply characterized by the fact that $\mathcal{Y}$ is a finite set, then each possible value of $y \in \mathcal{Y}$ characterizes a different class. For simplicity, we assume here that $\mathcal{Y} = \{1, \ldots, k\}$ where $k$ is the number of classes and denote for all $\ell \in [k]$:

$$\gamma_\ell := \mathbb{P}(Y = \ell); \qquad \mu_\ell := \mathbb{E}[x \mid y = \ell];$$
$$C_\ell := \mathbb{E}[xx^\top \mid y = \ell] - \mu_\ell \mu_\ell^\top.$$

Some natural extensions of Claim 4.1 to multiclass settings are already available in the literature for Gaussian data ((Mallory et al., 2025; Akhtiamov et al., 2024; Bosch & Panahi, 2025)). Assuming the claim is also valid in these cases, one can then deduce an extension of Theorem 4.4 to the multiclass settings as follows:

**Claim A.1** (Multiclass formulation of ERM asymptotics). *Under Assumptions 1–4, there exists some unique parameters $\mu_* \in \mathbb{R}^p$, $\alpha, \kappa, \nu \in \mathbb{R}_+^k$ such that $\forall \ell \in [k]$:*

$$Q(\nu) = \left(\sum_{h=1}^k \gamma_h \nu_h C_h + H\right)^{-1}, \qquad \kappa_\ell = \frac{1}{n}\mathrm{tr}\big(C_\ell Q(\nu)\big),$$

$$\nu_\ell = \frac{1}{\alpha_\ell \kappa_\ell}\mathbb{E}\big[z\,\xi_y\big(\mu_*^\top x + \alpha_\ell z, \kappa\big) \mid y = \ell\big]$$

$$\alpha_\ell^2 = \sum_{h=1}^k \frac{\gamma_h \mathrm{tr}\big(C_h Q(\nu) C_\ell Q(\nu)\big)}{\kappa_h^2 n}\mathbb{E}\Big[\xi_y\big(\mu_*^\top x + \alpha_h z, \kappa_h\big)^2 \mid y = h\Big]$$

$$\nabla\rho(\mu_*) + \sum_{h=1}^k \frac{\gamma_h}{\kappa_h}\mathbb{E}\big[x\,\xi_y\big(\mu_*^\top x + \alpha_h z, \kappa_h\big) \mid y = h\big] = 0.$$

*and for any bounded measurable function $f : \mathbb{R} \to \mathbb{R}$,*

$$\mathbb{E}\big[f(x^\top \hat{\theta}) \mid y = \ell\big] \xrightarrow[n\to\infty]{} \mathbb{E}\big[f(x^\top \mu_* + \alpha_\ell z) \mid y = \ell\big].$$

## B. Proofs of Subsection 3 results

For simplicity, in the next proposition we keep the notation $(X, Y)$ for deterministic objects. We define

$$F_{X,Y}(\theta) := \frac{1}{n}\sum_{i=1}^n \mathcal{L}_{y_i}(x_i^\top \theta) + \rho(\theta), \qquad \theta \in \mathbb{R}^p. \tag{16}$$

We will use multiple times the following classical stability lemma for minimizers.

**Lemma B.1** (Stability of minimizers under strong convexity). *Let $\phi, \psi : \mathbb{R}^p \to \mathbb{R}$ be convex and $\mathcal{C}^1$. Assume moreover that $\phi$ is $\mathcal{C}^2$ and $\kappa$-strongly convex, i.e.*

$$\forall x \in \mathbb{R}^p, \qquad \nabla^2\phi(x) \succeq \kappa I_p \quad \text{for some } \kappa > 0.$$

*Let $\mu_\phi := \arg\min_{x\in\mathbb{R}^p} \phi(x)$ (which is then unique). Then for any $x \in \mathbb{R}^p$,*

$$\|\mu_\phi - x\|_2 \le \frac{1}{\kappa}\|\nabla\phi(x)\|_2. \tag{17}$$

*In particular, if $\mu_\psi := \arg\min_{x\in\mathbb{R}^p} \psi(x)$ exists and $\psi$ is differentiable at $\mu_\psi$, then*

$$\|\mu_\phi - \mu_\psi\|_2 \le \frac{1}{\kappa}\|(\nabla\phi - \nabla\psi)(\mu_\psi)\|_2 \le \frac{1}{\kappa}\|\nabla\phi - \nabla\psi\|_{\mathcal{B}(0,\|\mu_\psi\|_2)}, \tag{18}$$

*where, for any $f : \mathbb{R}^p \to \mathbb{R}^p$ and any $C > 0$,*

$$\|f\|_{\mathcal{B}(0,C)} := \sup_{\|x\|_2 \le C}\|f(x)\|_2.$$

In words, Lemma B.1 bounds the distance between minimizers via gradients, provided one of the objectives is uniformly strongly convex and one has a radius bound on the other minimizer.

*Proof.* Since $\nabla^2 \phi(x) \succeq \kappa I_p$ for all $x$, the gradient map $\nabla \phi$ is $\kappa$-strongly monotone, i.e., for all $x, x' \in \mathbb{R}^p$,

$$\langle \nabla\phi(x) - \nabla\phi(x'), \, x - x' \rangle \geq \kappa \|x - x'\|_2^2.$$

By Cauchy–Schwarz,

$$\kappa \|x - x'\|_2^2 \leq \|\nabla\phi(x) - \nabla\phi(x')\|_2 \, \|x - x'\|_2,$$

hence (if $x \neq x'$; otherwise the inequality is trivial)

$$\|x - x'\|_2 \leq \frac{1}{\kappa} \, \|\nabla\phi(x) - \nabla\phi(x')\|_2 \, .$$

Taking $x' = \mu_\phi$ and using $\nabla\phi(\mu_\phi) = 0$ yields (17). If additionally $\mu_\psi$ is a minimizer of $\psi$ and $\psi$ is differentiable at $\mu_\psi$ (so $\nabla\psi(\mu_\psi) = 0$), then applying (17) with $x = \mu_\psi$ gives

$$\|\mu_\phi - \mu_\psi\|_2 \leq \frac{1}{\kappa} \|\nabla\phi(\mu_\psi)\|_2 = \frac{1}{\kappa} \|(\nabla\phi - \nabla\psi)(\mu_\psi)\|_2,$$

and the ball-sup bound follows immediately. $\qquad\square$

**Proposition B.2** (Lipschitz dependence of $\hat{\theta}$ on $(X, Y)$)**.** *For $(X, Y) \in \mathbb{R}^{p \times n} \times \mathcal{Y}^n$, define*

$$\hat{\theta}(X, Y) := \arg\min_{\theta \in \mathbb{R}^p} F_{X,Y}(\theta).$$

*Under Assumption 2, assume that there exists a constant $C > 0$ such that for all admissible $X, X'$,*

$$\|X\|_{\mathrm{op}}, \, \|X'\|_{\mathrm{op}} \leq C\sqrt{n}.$$

*Then*

$$\|\hat{\theta}(X, Y) - \hat{\theta}(X', Y')\|_2 \; \leq \; \frac{L}{\sqrt{n}} \, d((X, Y), (X', Y')),$$

*for some constant $L$ independent of $n$ and $p$.*

*Proof.* Since each $\mathcal{L}_{y_i}$ is convex and $\mathcal{C}^2$, we have $\mathcal{L}_{y_i}'' \geq 0$. Since $\mathcal{L}_{y_i}'$ is $\lambda$-Lipschitz, we also have $\mathcal{L}_{y_i}'' \leq \lambda$. Hence for all $\theta \in \mathbb{R}^p$,

$$\nabla^2 F_{X,Y}(\theta) = \frac{1}{n} X \operatorname{diag}\big(\mathcal{L}_{y_1}''(x_1^\top \theta), \ldots, \mathcal{L}_{y_n}''(x_n^\top \theta)\big) X^\top + \nabla^2 \rho(\theta) \succeq \kappa I_p, \tag{19}$$

so $F_{X,Y}$ is $\kappa$-strongly convex and admits a unique minimizer $\hat{\theta}(X, Y)$, characterized by $\nabla F_{X,Y}(\hat{\theta}(X, Y)) = 0$.

**Step 1: a uniform bound on $\|\hat{\theta}(X, Y)\|_2$.**  Applying (17) (Lemma B.1) to $\phi = F_{X,Y}$ at the point $x = 0$ yields

$$\kappa \|\hat{\theta}(X, Y)\|_2 \leq \|\nabla F_{X,Y}(0)\|_2 \, .$$

Let $f : \mathbb{R}^n \to \mathbb{R}^n$ be defined by $f(u)_i := \mathcal{L}_{y_i}'(u_i)$. Introduce the constants

$$B := \frac{1}{\sqrt{n}} \|f(0)\|_2, \qquad R := \|\nabla\rho(0)\|_2.$$

Then

$$\|\nabla F_{X,Y}(0)\|_2 = \left\| \frac{1}{n} X f(0) + \nabla\rho(0) \right\|_2 \leq \frac{1}{n} \|X\|_{\mathrm{op}} \|f(0)\|_2 + \|\nabla\rho(0)\|_2 \leq \frac{C\sqrt{n}}{n} \cdot \sqrt{n} B + R = CB + R.$$

Therefore,

$$\|\hat{\theta}(X, Y)\|_2 \leq \frac{1}{\kappa} \Big( CB + R \Big) =: M. \tag{20}$$

**Step 2: stability of minimizers under perturbations of** $(X, Y)$. Let $\theta := \hat{\theta}(X, Y)$ and $\theta' := \hat{\theta}(X', Y')$. Applying (18) (Lemma B.1) with $\phi = F_{X,Y}$ and $\psi = F_{X',Y'}$ gives

$$\|\theta - \theta'\|_2 \leq \frac{1}{\kappa}\|\nabla F_{X,Y} - \nabla F_{X',Y'}\|_{\mathcal{B}(0,\|\theta'\|_2)} \leq \frac{1}{\kappa}\|\nabla F_{X,Y} - \nabla F_{X',Y'}\|_{\mathcal{B}(0,M)}, \tag{21}$$

since $\|\theta'\|_2 \leq M$ by (20) applied to $(X', Y')$.

Fix an arbitrary $\bar{\theta} \in \mathbb{R}^p$ with $\|\bar{\theta}\|_2 \leq M$. We first bound the variation in $X$ at fixed $Y$:

$$\nabla F_{X,Y}(\bar{\theta}) - \nabla F_{X',Y}(\bar{\theta}) = \frac{1}{n}\Big(Xf(X^\top\bar{\theta}) - X'f(X'^\top\bar{\theta})\Big) = \frac{1}{n}\Big((X - X')f(X^\top\bar{\theta}) + X'\big(f(X^\top\bar{\theta}) - f(X'^\top\bar{\theta})\big)\Big).$$

Hence, using $\|X'\|_{\mathrm{op}} \leq C\sqrt{n}$,

$$\|\nabla F_{X,Y}(\bar{\theta}) - \nabla F_{X',Y}(\bar{\theta})\|_2 \leq \frac{1}{n}\|X - X'\|_{\mathrm{op}}\|f(X^\top\bar{\theta})\|_2 + \frac{1}{n}\|X'\|_{\mathrm{op}}\|f(X^\top\bar{\theta}) - f(X'^\top\bar{\theta})\|_2.$$

Since each coordinate $\mathcal{L}'_{y_i}$ is $\lambda$-Lipschitz, $f$ is $\lambda$-Lipschitz on $(\mathbb{R}^n, \|\cdot\|_2)$, hence

$$\|f(X^\top\bar{\theta}) - f(X'^\top\bar{\theta})\|_2 \leq \lambda\|(X - X')^\top\bar{\theta}\|_2 \leq \lambda\|X - X'\|_{\mathrm{op}}\|\bar{\theta}\|_2 \leq \lambda M\|X - X'\|_{\mathrm{op}}.$$

Also,

$$\|f(X^\top\bar{\theta})\|_2 \leq \|f(0)\|_2 + \lambda\|X^\top\bar{\theta}\|_2 \leq \sqrt{n}B + \lambda\|X\|_{\mathrm{op}}\|\bar{\theta}\|_2 \leq \sqrt{n}B + \lambda C\sqrt{n}\,M.$$

Combining,

$$\|\nabla F_{X,Y}(\bar{\theta}) - \nabla F_{X',Y}(\bar{\theta})\|_2 \leq \frac{1}{\sqrt{n}}\|X - X'\|_{\mathrm{op}}\Big(B + 2\lambda CM\Big). \tag{22}$$

Second, we bound the variation in $Y$ at fixed $X'$:

$$\|\nabla F_{X',Y}(\bar{\theta}) - \nabla F_{X',Y'}(\bar{\theta})\|_2 = \left\|\frac{1}{n}\sum_{i=1}^{n}\Big(\mathcal{L}'_{y_i}(x_i'^\top\bar{\theta}) - \mathcal{L}'_{y_i'}(x_i'^\top\bar{\theta})\Big)x_i'\right\|_2 \tag{23}$$

$$\leq \frac{1}{n}\|X'\|_{\mathrm{op}}\left(\sum_{i=1}^{n}\Big(\mathcal{L}'_{y_i}(x_i'^\top\bar{\theta}) - \mathcal{L}'_{y_i'}(x_i'^\top\bar{\theta})\Big)^2\right)^{1/2}. \tag{24}$$

Under the regularity assumption that $y \mapsto \mathcal{L}'_y(t)$ is $\lambda$-Lipschitz uniformly in $t$ (with respect to $d_{\mathcal{Y}}$),

$$\left|\mathcal{L}'_{y_i}(t) - \mathcal{L}'_{y_i'}(t)\right| \leq \lambda\, d_{\mathcal{Y}}(y_i, y_i').$$

Using $\|X'\|_{\mathrm{op}} \leq C\sqrt{n}$ in (23) yields

$$\|\nabla F_{X',Y}(\bar{\theta}) - \nabla F_{X',Y'}(\bar{\theta})\|_2 \leq \frac{\lambda C}{\sqrt{n}}\, d_{\mathcal{Y}}(Y, Y'), \qquad d_{\mathcal{Y}}(Y, Y') := \left(\sum_{i=1}^{n} d_{\mathcal{Y}}(y_i, y_i')^2\right)^{1/2}.$$

Finally, combining the $X$ and $Y$ variations, inserting into (21), and using (20) gives

$$\|\hat{\theta}(X, Y) - \hat{\theta}(X', Y')\|_2 \leq \frac{1}{\kappa\sqrt{n}}\left(\Big(B + 2\lambda CM\Big)\|X - X'\|_{\mathrm{op}} + \lambda C\, d_{\mathcal{Y}}(Y, Y')\right) \leq \frac{L}{\sqrt{n}}\, d\big((X, Y), (X', Y')\big),$$

for a constant $L$ independent of $n, p$ (absorbing $\kappa, C, B, M, \lambda$ into $L$ and using the definition of $d$). $\qquad\square$

*Proof of Theorem 2.1 – Concentration of $\hat{\theta}$.* This is an application of the Lipschitz property from Proposition B.2 together with Assumption 1 and Assumption 2. $\qquad\square$

*Proof of Proposition 3.1 – quadratic approximation of $\rho$.* Let $\mu_{\hat{\theta}} := \mathbb{E}[\hat{\theta}]$ and $\theta_0 := \hat{\theta} - \mu_{\hat{\theta}}$. A Taylor expansion of $\rho$ at $\mu_{\hat{\theta}}$ with integral (Taylor–Lagrange) remainder yields

$$\left| \rho(\mu_{\hat{\theta}} + \theta_0) - \rho(\mu_{\hat{\theta}}) - \nabla\rho(\mu_{\hat{\theta}})^\top \theta_0 - \frac{1}{2}\theta_0^\top \nabla^2\rho(\mu_{\hat{\theta}})\theta_0 \right| \leq \frac{1}{6}\sup_{t\in[0,1]} \left| \nabla^3\rho(\mu_{\hat{\theta}} + t\theta_0) \cdot (\theta_0, \theta_0, \theta_0) \right|. \tag{25}$$

Using the decomposition of the third-order tensor into diagonal and off-diagonal parts, and the bound (valid for any $\theta, u \in \mathbb{R}^p$)

$$\left| \nabla^3\rho(\theta) \cdot (u, u, u) \right| \leq \|\operatorname{Diag}(\nabla^3\rho(\theta))\|_F \|u^{(3)}\| + \|\operatorname{OffDiag}(\nabla^3\rho(\theta))\|_{\mathrm{op}} \|u\|^3 \tag{26}$$

$$\leq C\sqrt{n}\,\|u^{(3)}\| + \frac{1}{\sqrt{n}}\,\|u\|^3, \tag{27}$$

(where we used $\|\operatorname{Diag}(\nabla^3\rho(\theta))\|_F \leq \sqrt{p}\,\|\nabla^3\rho(\theta)\|_{\mathrm{op}} \leq C\sqrt{n}$ under Assumption 3), we deduce from (25) that

$$\left| \rho(\hat{\theta}) - \rho(\mu_{\hat{\theta}}) - \nabla\rho(\mu_{\hat{\theta}})^\top \theta_0 - \frac{1}{2}\theta_0^\top \nabla^2\rho(\mu_{\hat{\theta}})\theta_0 \right| \leq C\max\left( \sqrt{n}\|\theta_0^{(3)}\|,\ \frac{1}{\sqrt{n}}\|\theta_0\|^3 \right). \tag{28}$$

Relying on Theorem 2.1 and (Louart, 2023, Example B.38 and Proposition B.22), we can bound, for some $C, c > 0$,

$$\mathbb{P}(\|\theta_0\| \geq t) \leq Ce^{-ct^2}, \qquad \mathbb{P}(\|\theta_0^{(3)}\| \geq t) \leq Ce^{-cn^2t^2/\log n} + Ce^{-cnt} + Ce^{-cnt^{2/3}}.$$

Combining with (28) yields the announced concentration bound. □

**Lemma B.3.** *Given a tensor $T \in \mathbb{R}^{p\times p\times p}$ and $u \in \mathbb{R}^p$, the contraction $T \cdot u \in \mathbb{R}^{p\times p}$ satisfies*

$$\|T \cdot u\|_F \leq \sqrt{p}\,\|T \cdot u\|_{\mathrm{op}} \leq \sqrt{p}\,\|T\|_{\mathrm{op}}\|u\|.$$

*Proof of Lemma 3.2 – Constant approximation of $\nabla^2\rho$.* A Taylor–Lagrange argument for the Hessian gives

$$\left\| \nabla^2\rho(\theta) - \nabla^2\rho(0) \right\|_F \leq \sup_{t\in[0,1]} \left\| \nabla^3\rho(t\theta) \cdot \theta \right\|_F. \tag{29}$$

Using Lemma B.3 and the decomposition into diagonal/off-diagonal terms,

$$\left\| \nabla^3\rho(t\theta) \cdot \theta \right\|_F \leq \left\| \operatorname{Diag}(\nabla^3\rho(t\theta)) \cdot \theta \right\|_F + \left\| \operatorname{OffDiag}(\nabla^3\rho(t\theta)) \cdot \theta \right\|_F$$
$$\leq \|\operatorname{Diag}(\nabla^3\rho(t\theta))\|_{\mathrm{op}} \|\theta\| + \sqrt{p}\,\|\operatorname{OffDiag}(\nabla^3\rho(t\theta))\|_{\mathrm{op}} \|\theta\| \leq C\|\theta\|,$$

by Assumption 3. Plugging into (29) yields

$$\|\nabla^2\rho(\theta) - \nabla^2\rho(0)\|_F \leq C\|\theta\|.$$

□

# C. Proofs of Subsection 4 results

Let us first introduce a result that explains where the resolvent comes from.

**Lemma C.1** (Quadratic minimization on an ellipsoid: dual representation). *Let $p \geq 1$, let $H, C \in \mathbb{S}^p_{++}$ and $u \in \mathbb{R}^p$, and let $\alpha \geq 0$. Define the constrained quadratic program*

$$p(\alpha) := \min_{\theta\in\mathbb{R}^p} \left\{ w^\top\theta + \frac{1}{2}\theta^\top H\theta\ :\ \theta^\top C\theta = \alpha^2 \right\}. \tag{30}$$

*Set the domain*

$$\mathcal{D} := \{\nu \in \mathbb{R} :\ H + \nu C \succ 0\},$$

*and for $\nu \in \mathcal{D}$ define*

$$d(\nu) := -\frac{1}{2}w^\top(H + \nu C)^{-1}w - \frac{\alpha^2\nu}{2}. \tag{31}$$

*Then*

$$p(\alpha) = \sup_{\nu\in\mathcal{D}} d(\nu). \tag{32}$$

*Proof.* The claim is trivial when $\alpha = 0$ since the constraint forces $\theta = 0$, hence $p(0) = 0$, and also $\sup_{\nu \in \mathcal{D}} d(\nu) = 0$ (take $\nu \to +\infty$ in (31)).

Assume $\alpha > 0$ in the remainder. Let $A := C^{-1/2}HC^{-1/2} \in \mathbb{S}^p_{++}$ and $\tilde{w} := C^{-1/2}w$. With the change of variables $\theta = \alpha C^{-1/2}z$, the constraint becomes $\|z\|^2 = 1$ and

$$w^\top \theta + \frac{1}{2}\theta^\top H\theta = \alpha\,\tilde{w}^\top z + \frac{\alpha^2}{2}\,z^\top Az.$$

Therefore,

$$p(\alpha) = \min_{\|z\|=1}\left\{\alpha\,\tilde{w}^\top z + \frac{\alpha^2}{2}\,z^\top Az\right\}. \tag{33}$$

For $\lambda \in \mathbb{R}$, consider the Lagrangian

$$\mathcal{L}(z,\lambda) = \alpha\,\tilde{w}^\top z + \frac{\alpha^2}{2}\,z^\top Az + \frac{\lambda}{2}(\|z\|^2 - 1).$$

For any $\lambda$ and any feasible $z$ (i.e., $\|z\| = 1$) we have $\mathcal{L}(z,\lambda) = \alpha\,\tilde{w}^\top z + \frac{\alpha^2}{2}z^\top Az$, hence by taking the infimum over $z \in \mathbb{R}^p$,

$$\inf_{z \in \mathbb{R}^p} \mathcal{L}(z,\lambda) \leq p(\alpha). \tag{34}$$

If $\alpha^2 A + \lambda I \succ 0$, the function $z \mapsto \mathcal{L}(z,\lambda)$ is strictly convex and its unique minimizer satisfies

$$\nabla_z \mathcal{L}(z,\lambda) = \alpha\tilde{w} + (\alpha^2 A + \lambda I)z = 0 \quad \Longrightarrow \quad z(\lambda) = -(\alpha^2 A + \lambda I)^{-1}\alpha\tilde{w}.$$

Substituting back yields

$$\inf_z \mathcal{L}(z,\lambda) = -\frac{\alpha^2}{2}\,\tilde{w}^\top(\alpha^2 A + \lambda I)^{-1}\tilde{w} - \frac{\lambda}{2}, \qquad \text{for } \alpha^2 A + \lambda I \succ 0. \tag{35}$$

Since the feasible set $\{z : \|z\| = 1\}$ is nonempty and the objective in (33) is continuous, a minimizer $z^\star$ exists. The constraint is smooth and $\nabla(\|z\|^2 - 1) = 2z^\star \neq 0$ on the sphere, hence the Lagrange multiplier theorem gives some $\lambda^\star \in \mathbb{R}$ such that

$$\alpha\tilde{w} + \alpha^2 Az^\star + \lambda^\star z^\star = 0.$$

If $\alpha^2 A + \lambda I$ has a negative eigenvalue, then $\inf_z \mathcal{L}(z,\lambda) = -\infty$, so such $\lambda$ do not contribute to the dual supremum. Therefore it suffices to consider $\lambda$ such that $\alpha^2 A + \lambda I \succeq 0$, and we first treat the case $\succ 0$.

If $\alpha^2 A + \lambda^\star I \succ 0$, then $z^\star = z(\lambda^\star)$ and therefore

$$p(\alpha) = \mathcal{L}(z^\star,\lambda^\star) = \inf_z \mathcal{L}(z,\lambda^\star).$$

Combining with (34)–(35) yields

$$p(\alpha) = \sup_{\lambda:\ \alpha^2 A + \lambda I \succ 0}\left\{-\frac{\alpha^2}{2}\,\tilde{w}^\top(\alpha^2 A + \lambda I)^{-1}\tilde{w} - \frac{\lambda}{2}\right\}. \tag{36}$$

If instead $\alpha^2 A + \lambda^\star I \succeq 0$ is singular, the same identity holds with a supremum by a standard limiting argument (approximate $\lambda^\star$ from above so that $\alpha^2 A + \lambda I \succ 0$ and pass to the limit); this is why we state (32) with "sup".

Let $\nu := \lambda/\alpha^2$. Then $\alpha^2 A + \lambda I = \alpha^2(A + \nu I)$ and, using

$$H + \nu C = C^{1/2}(A + \nu I)C^{1/2}, \qquad (H + \nu C)^{-1} = C^{-1/2}(A + \nu I)^{-1}C^{-1/2},$$

we obtain

$$\tilde{w}^\top(A + \nu I)^{-1}\tilde{w} = w^\top(H + \nu C)^{-1}w.$$

Thus (36) becomes exactly (32) with $d(\nu)$ as in (31). $\qquad\square$

**Proposition C.2** (Resolution of the auxiliary problem)**.** *Let $\mu, x \in \mathbb{R}^p$, $h \in \mathbb{R}^n$, and let $C, H \in \mathbb{R}^{p \times p}$ be symmetric positive definite. Let $\mathcal{L} : \mathbb{R}^n \to (-\infty, +\infty]$ be proper, l.s.c. and convex, and assume that the saddle-point problem below admits a unique saddle point $(\tilde{\theta}, \tilde{w})$:*

$$(\tilde{\theta}, \tilde{w}) := \arg\min_{\theta \in \mathbb{R}^p} \max_{w \in \mathbb{R}^n} \left\{ \|C^{1/2}\theta\| \, w^\top h + (\mu^\top x) \, w^\top \mathbf{1} + \|w\| \, \theta^\top x - \frac{1}{n}\mathcal{L}^*(nw) + \tfrac{1}{2}\theta^\top H\theta \right\}. \tag{37}$$

*Denote $\tilde{\alpha} := \|C^{1/2}\tilde{\theta}\|$ and $\tilde{\beta} := \sqrt{n}\|\tilde{w}\|$. Then $(\tilde{\alpha}, \tilde{\kappa}, \tilde{\beta}, \tilde{\nu})$ can be characterized through the saddle point*

$$(\tilde{\alpha}, \tilde{\kappa}, \tilde{\beta}, \tilde{\nu}) = \arg\min_{\alpha \geq 0, \kappa > 0} \arg\max_{\beta \geq 0, \nu \in \mathcal{D}} \left\{ \frac{\beta^2 \kappa}{2} + \frac{1}{n}e_{\mathcal{L}}\big(\alpha g + (\mu^\top x)\mathbf{1}; \kappa\big) - \frac{\beta^2}{2n}x^\top(H + \nu C)^{-1}x - \frac{\alpha^2 \nu}{2} \right\}, \tag{38}$$

*where $\mathcal{D} := \{\nu \in \mathbb{R} : H + \nu C \succ 0\}$.*

*Proof.* Since $\mathcal{L}$ is proper l.s.c. convex, the Fenchel–Moreau theorem yields

$$\mathcal{L}^*(z) = \sup_{v \in \mathbb{R}^n} \{v^\top z - \mathcal{L}(v)\} \quad \Longrightarrow \quad -\frac{1}{n}\mathcal{L}^*(nw) = \min_{v \in \mathbb{R}^n} \left\{ -w^\top v + \frac{1}{n}\mathcal{L}(v) \right\}.$$

We thus have the identity:

$$(\tilde{\theta}, \tilde{w}, \sim) = \arg\min_{\theta} \max_{w} \min_{v} \left\{ \|C^{1/2}\theta\| \, w^\top h + (\mu^\top x) \, w^\top \mathbf{1} - w^\top v + \|w\| \, \theta^\top x + \frac{1}{n}\mathcal{L}(v) + \tfrac{1}{2}\theta^\top H\theta \right\}. \tag{39}$$

Write $w = (\beta/\sqrt{n})u$ with $\beta := \sqrt{n}\|w\| \geq 0$ and $u \in \mathbb{S}^{n-1}$. For fixed $(\theta, \beta)$, consider the inner game in $(u, v)$:

$$\phi_s(u, v) := \frac{\beta}{\sqrt{n}}u^\top \big( \|C^{1/2}\theta\| \, h + (\mu^\top x)\mathbf{1} - v \big) + \frac{1}{n}\mathcal{L}(v).$$

Then, recalling that $\tilde{\beta} = \sqrt{n}\|\tilde{w}\|$,

$$(\tilde{\theta}, \tilde{\beta}, \sim, \sim) = \arg\min_{\theta} \max_{\beta \geq 0} \max_{u \in \mathbb{S}^{n-1}} \min_{v} \left\{ \phi_s(u, v) + \frac{\beta}{\sqrt{n}}\theta^\top x + \tfrac{1}{2}\theta^\top H\theta \right\}$$

$$= \arg\min_{\theta} \max_{\beta \geq 0} \max_{u \in \mathcal{B}(0,1)} \min_{v} \left\{ \phi_s(u, v) + \frac{\beta}{\sqrt{n}}\theta^\top x + \tfrac{1}{2}\theta^\top H\theta \right\},$$

since $\phi_s$ is affine in $u$ and its maximum over the sphere equals its maximum over the unit ball.

The map $(u, v) \mapsto \phi_s(u, v)$ is concave in $u$ on the compact convex set $\mathcal{B}(0, 1)$ and convex l.s.c. in $v$ on $\mathbb{R}^n$. Therefore, by Sion's minimax theorem,

$$\max_{u \in \mathcal{B}(0,1)} \min_{v \in \mathbb{R}^n} \phi_s(u, v) = \min_{v \in \mathbb{R}^n} \max_{u \in \mathcal{B}(0,1)} \phi_s(u, v)$$

$$= \min_{v \in \mathbb{R}^n} \left\{ \frac{\beta}{\sqrt{n}}\big\| \|C^{1/2}\theta\| \, h + (\mu^\top x)\mathbf{1} - v \big\| + \frac{1}{n}\mathcal{L}(v) \right\}.$$

Plugging this into (39) yields

$$(\tilde{\theta}, \tilde{\beta}) = \arg\min_{\theta} \max_{\beta \geq 0} \left\{ \phi_{\mathcal{L}}(\|C^{1/2}\theta\|, \beta) + \frac{\beta}{\sqrt{n}}\theta^\top x + \tfrac{1}{2}\theta^\top H\theta \right\}, \tag{40}$$

where, for $\alpha \geq 0$ and $\beta \geq 0$,

$$\phi_{\mathcal{L}}(\alpha, \beta) := \min_{v \in \mathbb{R}^n} \left\{ \frac{\beta}{\sqrt{n}}\big\| \alpha g + (\mu^\top x)\mathbf{1} - v \big\| + \frac{1}{n}\mathcal{L}(v) \right\}.$$

Let $u := \alpha g + (\mu^\top x)\mathbf{1}$. Using the standard identity $t = \min_{\tau>0}\{\tau/2 + t^2/(2\tau)\}$ for $t \geq 0$, we obtain

$$\phi_{\mathcal{L}}(\alpha, \beta) = \min_v \left\{ \frac{\beta}{\sqrt{n}}\|u - v\| + \frac{1}{n}\mathcal{L}(v) \right\}$$

$$= \min_{\tau_{\mathcal{L}}>0,v} \left\{ \frac{\beta\tau_{\mathcal{L}}}{2} + \frac{\beta}{2\tau_{\mathcal{L}}n}\|u - v\|^2 + \frac{1}{n}\mathcal{L}(v) \right\}.$$

Set $\kappa := \tau_{\mathcal{L}}/\beta$ (with the convention that $\kappa$ is optimized only when $\beta > 0$). Then $\beta\tau_{\mathcal{L}}/2 = \beta^2\kappa/2$ and $\beta/(2\tau_{\mathcal{L}}) = 1/(2\kappa)$, hence

$$\phi_{\mathcal{L}}(\alpha, \beta) = \min_{\kappa>0} \left\{ \frac{\beta^2\kappa}{2} + \frac{1}{n}e_{\mathcal{L}}(u; \kappa) \right\}, \qquad e_{\mathcal{L}}(u; \kappa) := \min_{v \in \mathbb{R}^n} \left\{ \frac{1}{2\kappa}\|u - v\|^2 + \mathcal{L}(v) \right\}. \tag{41}$$

Injecting (41) into (40) gives

$$(\tilde{\theta}, \tilde{\beta}, \sim) = \arg\min_\theta \max_{\beta\geq 0} \min_{\kappa>0} \left\{ \frac{\beta^2\kappa}{2} + \frac{1}{n}e_{\mathcal{L}}\Big(\|C^{1/2}\theta\|\, h + (\mu^\top x)\mathbf{1}; \kappa\Big) + \frac{\beta}{\sqrt{n}}\theta^\top x + \tfrac{1}{2}\theta^\top H\theta \right\}. \tag{42}$$

Let $\alpha := \|C^{1/2}\theta\| \geq 0$. For fixed $(\alpha, \beta, \kappa)$, the dependence in $\theta$ inside (42) is through

$$\frac{\beta}{\sqrt{n}}x^\top\theta + \tfrac{1}{2}\theta^\top H\theta \quad \text{under the constraint} \quad \theta^\top C\theta = \alpha^2.$$

Lemma C.1 yields

$$\min_{\theta:\, \theta^\top C\theta = \alpha^2} \left\{ \frac{\beta}{\sqrt{n}}x^\top\theta + \tfrac{1}{2}\theta^\top H\theta \right\} = \max_{\nu \in \mathcal{D}} \left\{ -\frac{\beta^2}{2n}x^\top(H + \nu C)^{-1}x - \frac{\alpha^2\nu}{2} \right\}.$$

Combining this with (42) and using that the remaining term depends on $\theta$ only through $\alpha$, we obtain the claimed characterization (38). □

**Lemma C.3.** *Let $E$ be a finite-dimensional real vector space, let $Z$ be a random variable with values in $\mathcal{Z}$, and for each $z \in \mathcal{Z}$ let $\psi_z : E \to \mathbb{R}$ be $\mathcal{C}^1$, strictly convex, and admitting a unique minimizer. Define the random minimizer*

$$\hat{u} := \arg\min_{u \in E} \psi_Z(u).$$

*Assume that $\hat{u}$ is integrable and that for all $m \in E$ the quantity $\mathbb{E}[\psi_Z(\hat{u} - \mathbb{E}[\hat{u}] + m)]$ is finite. Then $\mathbb{E}[\hat{u}]$ is the (unique) minimizer of*

$$m \longmapsto \mathbb{E}\big[\psi_Z\big((\hat{u} - \mathbb{E}[\hat{u}]) + m\big)\big],$$

*i.e.*

$$\mathbb{E}[\hat{u}] = \arg\min_{m \in E} \mathbb{E}\big[\psi_Z\big((\hat{u} - \mathbb{E}[\hat{u}]) + m\big)\big].$$

*Proof.* For each realization $Z = z$, the point $\hat{u} = \hat{u}(z)$ minimizes $\psi_z$, hence $\nabla\psi_z(\hat{u}) = 0$. By convexity, for any $m \in E$,

$$\psi_z\big(\hat{u} - \mathbb{E}[\hat{u}] + m\big) \geq \psi_z(\hat{u}) + \langle\nabla\psi_z(\hat{u}),\, m - \mathbb{E}[\hat{u}]\rangle = \psi_z(\hat{u}).$$

Taking expectations yields

$$\mathbb{E}\big[\psi_Z\big(\hat{u} - \mathbb{E}[\hat{u}] + m\big)\big] \geq \mathbb{E}[\psi_Z(\hat{u})] = \mathbb{E}\big[\psi_Z\big(\hat{u} - \mathbb{E}[\hat{u}] + \mathbb{E}[\hat{u}]\big)\big],$$

so $m = \mathbb{E}[\hat{u}]$ is a minimizer. Uniqueness follows from strict convexity. □

**Corollary C.4.** *Under Assumption 2, letting $\hat{\theta}_0 = \hat{\theta} - \mu_{\hat{\theta}}$, we have*

$$\mu_{\hat{\theta}} = \arg\min_{\mu \in \mathbb{R}^p} \mathbb{E}\left[\frac{1}{n}\sum_{i=1}^n \mathcal{L}_{y_i}\big(x_i^\top(\hat{\theta}_0 + \mu)\big) + \rho(\hat{\theta}_0 + \mu)\right].$$

**Proposition C.5.** *Under Assumption 1 and Assumption 2, the parameter $\tilde{\alpha}$ defined in (38) satisfies*

$$\mathbb{P}\left(|\tilde{\alpha} - \alpha_*| \geq t\right) \leq Ce^{-cnt^2},$$

*for constants $C, c > 0$ independent of $n, p$, where $\alpha_*$ is defined as*

$$(\alpha_*, \sim, \sim, \sim) = \arg\min_{\alpha \geq 0, \kappa > 0} \arg\max_{\beta \geq 0, \nu \in \mathcal{D}} \left\{ \frac{\beta^2 \kappa}{2} + \frac{1}{n}\mathbb{E}\left[e_{\mathcal{L}}\left(\alpha g + (\mu^\top x)\mathbf{1}; \kappa\right)\right] - \frac{\beta^2}{2n}\text{tr}\left(C_x\left(H + \nu C\right)^{-1}\right) - \frac{\alpha^2 \nu}{2} \right\}.$$

*Proof.* Let us introduce the mapping

$$f : (\alpha, \nu, \beta, \kappa) \mapsto \frac{\beta^2 \kappa}{2} + \frac{1}{n}e_{\mathcal{L}}\left(\alpha g + (\mu^\top x)\mathbf{1}; \kappa\right) - \frac{\beta^2}{2n} x^\top (H + \nu C)^{-1}x - \frac{\alpha^2 \nu}{2}.$$

This objective is (typically) strictly convex in $\alpha$. As in the proof of Proposition B.2, one can rely on Lemma B.1 to show concentration of $\tilde{\alpha}$ after bounding the variations of the $\alpha$-gradient

$$\frac{\partial f}{\partial \alpha} = \frac{1}{n}\left\langle \nabla_u e_{\mathcal{L}}(\alpha g + (\mu^\top x)\mathbf{1}; \kappa), h \right\rangle - \alpha \nu,$$

with respect to fluctuations of $h$ (and $x$ when applicable). Under the regularity properties of the Moreau envelope (bounded first and second derivatives under Assumption 2), this yields that $\tilde{\alpha}$ is bounded and is a $C/\sqrt{n}$-Lipschitz function of $h$ (details omitted), hence

$$\mathbb{P}\left(|\tilde{\alpha} - \mathbb{E}[\tilde{\alpha}]| \geq t\right) \leq Ce^{-cnt^2}, \tag{43}$$

for some constants $C, c > 0$.

Denote $\tilde{\alpha}_0 := \tilde{\alpha} - \mathbb{E}[\tilde{\alpha}]$. By Lemma C.3, $\mathbb{E}[\tilde{\alpha}]$ can be characterized as the minimizer of the expected shifted objective (obtained by replacing $\alpha$ with $\mu_\alpha + \tilde{\alpha}_0$ in $f$). Comparing this expected objective with the deterministic limit defining $\alpha_*$ and using again Lemma B.1 (together with the Lipschitz properties of the $\alpha$-gradient) yields $|\mathbb{E}[\tilde{\alpha}] - \alpha_*| \leq C/\sqrt{n}$. Combining this bias bound with (43) concludes the proof. □

*Proof of Theorem 4.3 – Minmax formulation.* Applying Claim 4.1 to (10), we obtain that for any $\varepsilon > 0$,

$$\mathbb{P}\left(\left|\|C^{\frac{1}{2}}\hat{\theta}_0\| - \mathbb{E}[\|C^{\frac{1}{2}}\check{\theta}_0\|]\right| \geq \varepsilon\right) \xrightarrow[n \to \infty]{} 0,$$

where $(\check{\theta}, \check{w})$ is defined by

$$(\check{\theta}, \check{w}) := \arg\min_{\theta \in \mathbb{R}^p} \max_{w \in \mathbb{R}^n} \left\{ \|C^{1/2}\theta\|\, w^\top g + (\mu_{\hat{\theta}}^\top x)\, w^\top \mathbf{1} + \|w\|\, \theta^\top x - \frac{1}{n}\mathcal{L}^*(nw) + \rho(\theta + \mu_{\hat{\theta}}) \right\}.$$

This is the same minimization problem as in Proposition C.2, up to replacing $\frac{1}{2}\theta^\top H\theta$ with $\rho(\theta + \mu_{\hat{\theta}})$ (constants such as $\rho(\mu_{\hat{\theta}})$ do not affect the minimization).

Using Proposition 3.1 to approximate $\rho(\theta + \mu_{\hat{\theta}})$ by its quadratic expansion and invoking stability of the saddle point (details omitted), Proposition C.2 yields

$$\mathbb{P}\left(\left|\|C^{\frac{1}{2}}\hat{\theta}_0\| - \mathbb{E}[\tilde{\alpha}]\right| \geq \varepsilon\right) \xrightarrow[n \to \infty]{} 0.$$

Finally, Proposition C.5 gives

$$\mathbb{P}\left(\left|\|C^{\frac{1}{2}}\hat{\theta}_0\| - \alpha_*\right| \geq \varepsilon\right) \xrightarrow[n \to \infty]{} 0.$$

To set that $\mu_{\hat{\theta}} \xrightarrow[n \to \infty]{} \mu_*$, we will rely on the fact that:

- thanks to Corollary C.4:

$$\mu_{\hat{\theta}} = \arg\min_{\mu \in \mathbb{R}^p} \mathbb{E}\left[\Phi_\mu(X)\right] \qquad \text{with } \Phi_\mu(X) := \frac{1}{n}\sum_{i=1}^n \mathcal{L}_{y_i}\left(x_i^\top(\hat{\theta}_0 + \mu)\right) + \rho(\hat{\theta}_0 + \mu)$$

- by definition (6):

$$\mu_* = \arg\min_{\mu \in \mathbb{R}^p} \mathbb{E}\left[\phi_\mu(x,g)\right] \quad \text{with} \ \ \phi_\mu(x,g) := \frac{\beta^2 \kappa}{2} + \frac{1}{n} e_{\mathcal{L}}\left(\alpha g + (\mu^\top x)\mathbf{1}; \kappa\right) - \frac{\beta^2}{2n} x^\top (H + \nu C)^{-1} x - \frac{\alpha^2 \nu}{2}$$

Now, we know from the first result of Claim 4.1 that:

$$\forall \varepsilon > 0 : \qquad \mathbb{P}\left(|\Phi_\mu(X) - \phi_\mu(x,g)| \geq \varepsilon\right) \xrightarrow[n\to\infty]{} 0,$$

which directly implies:

$$|\mathbb{E}[\Phi_\mu(X)] - \mathbb{E}[\phi_\mu(x,g)]| \xrightarrow[n\to\infty]{} 0$$

Now, the mapping $\mu \mapsto [\Phi_\mu(X)]$ being $\kappa$-strictly convex, one can either look at the derivative of $\mu$ and employ Lemma B.1, either rely on a classical result of variational analysis, e.g., (Rockafellar & Wets, 1998, Theorem 7.33) that allows us to deduce convergence of minimizers from convergence of objectives to get:

$$\|\mu_{\hat{\theta}} - \mu_*\| \xrightarrow[n\to\infty]{} 0.$$

$\square$

*Proof of Theorem 4.4 – Fixed-point formulation.* Basic properties of the Moreau envelope yield

$$\partial_u e_{\mathcal{L}_y} \xi_y(u,\kappa) = \frac{1}{\kappa}(u,\kappa), \qquad \partial_\kappa e_{\mathcal{L}_y}(u,\kappa) = -\frac{1}{2\kappa^2} \xi_y(u,\kappa)^2.$$

Treat $\mu, \alpha, \kappa$ as minimization variables and $\beta, \nu$ as maximization variables in (4) and denote $u := x^\top \mu + \alpha z$. The stationarity conditions are:

- Gradient with respect to $\mu$:

$$\nabla \rho(\mu) + \frac{1}{\kappa}\mathbb{E}[\xi_y(u,\kappa)\,x] = 0.$$

- Gradient with respect to $\alpha$:

$$\frac{1}{\kappa}\mathbb{E}[\xi_y(u,\kappa)\,z] - \nu\alpha = 0 \quad \Longrightarrow \quad \nu\alpha = \frac{1}{\kappa}\mathbb{E}[rz].$$

- Gradient with respect to $\kappa$:

$$\frac{\beta^2}{2} - \frac{1}{2\kappa^2}\mathbb{E}[\xi_y(u,\kappa)^2] = 0 \quad \Longrightarrow \quad \beta^2 = \frac{\mathbb{E}[r^2]}{\kappa^2}.$$

- Gradient with respect to $\beta$:

$$\beta\left(\kappa - \frac{1}{n}\mathrm{tr}(C_x Q(\nu))\right) = 0 \quad \Longrightarrow \quad \kappa = \frac{1}{n}\mathrm{tr}(C_x Q(\nu)),$$

since $\beta > 0$ at the optimum.

- Gradient with respect to $\nu$. Using $\frac{d}{d\nu}Q(\nu) = -Q(\nu)C_x Q(\nu)$, we have

$$\frac{d}{d\nu}\mathrm{tr}(C_x Q(\nu)) = -\mathrm{tr}\big(C_x Q(\nu) C_x Q(\nu)\big) = -nA(\nu),$$

with

$$A(\nu) := \frac{1}{n}\mathrm{tr}\big(C_x Q(\nu) C_x Q(\nu)\big).$$

Then stationarity gives

$$-\frac{\alpha^2}{2} - \frac{\beta^2}{2n}\frac{d}{d\nu}\mathrm{tr}(C_x Q(\nu)) = 0 \quad \Longrightarrow \quad \frac{\alpha^2}{\beta^2} = A(\nu).$$

Combining the last three identities to eliminate $\alpha$ and $\beta$ yields

$$\nu^2 A(\nu)\,\mathbb{E}[\xi_y(u,\kappa)^2] = \big(\mathbb{E}[\xi_y(u,\kappa)z]\big)^2.$$

$\square$

## D. Proofs of Section 5 results and Link with random matrix theory methods

### D.1. Proof of Corollary 5.1

Given $y \in \mathcal{Y}$, to set the close form expression of $\xi_y$ given in (14) let us merely note from elementary calculus that in the case of a square loss:

$$\operatorname{prox}_{\kappa \mathcal{L}_y}(u) = \frac{u + \kappa y}{1 + \kappa}$$

Denoting for any $\mu \in \mathbb{R}^p$, $\alpha \kappa > 0$:

$$u := \mu_\theta^\top x + \alpha z, \qquad \gamma := \frac{\kappa}{1 + \kappa} \qquad \text{and} \qquad \Delta := \mathbb{E}\big[((\mu_\theta - \theta^*)^\top x)^2\big] = (\mu_\theta - \theta^*)^\top \Sigma_x (\mu_\theta - \theta^*)$$

we get:

$$\mathbb{E}[\xi_y(u)\, z] = \gamma\, \alpha, \qquad \mathbb{E}[\xi_y(u)\, x] = \gamma\, C_x (\mu_\theta - \theta^*), \qquad \text{and} \qquad \mathbb{E}[\xi_y(u)^2] = \gamma^2 \Big(\Delta + \alpha^2 + \sigma_\varepsilon^2\Big) \tag{44}$$

Then, denoting $(\mu_*, \alpha_*, \kappa_*, \nu_*)$ the fixed point of the equation provided in Theorem 4.4, we get with the upper identities the formula:

$$\nu_* = \frac{1}{\kappa_* \alpha_*} \mathbb{E}\big[\xi_y(x^\top \mu_* + \alpha_* z)\, z\big] = \frac{1}{1 + \kappa_*} = \frac{1}{1 + \frac{1}{n} \operatorname{tr}\big(C_x\, (\nu_* C_x + H)^{-1}\big)},$$

which is exactly (15). The vector $\mu_*$ of Theorem 4.4 satisfies with this setting:

$$\nabla \rho(\mu_*) = -\frac{1}{\kappa} \mathbb{E}\big[x\, \xi_y\big(\mu_*^\top x + \alpha_* z, \kappa\big)\big] = -\nu C_x(\mu_* - \theta^*),$$

Replacing $\nabla \rho(\mu_*)$ with its value $a + H\mu_*$, that translates:

$$\mu_* = (H + \nu C_x)^{-1}\, (\nu C_x \theta^* - a)\,.$$

Then, recalling the notation $\Delta(\nu_*) = (\mu_* - \theta^*)^\top \Sigma_x (\mu_* - \theta^*)$, where we introduced for simplicity the notation $\Sigma_x := \mathbb{E}[xx^\top] = C_x + \mu_x \mu_x^\top$, one can apply Theorem 4.3 to get:

$$\mathbb{E}[\mathcal{L}_y(x^\top \hat{\theta})^2] = \mathbb{E}[(x^\top(\hat{\theta} - \theta^*))^2] = (\mu_{\hat{\theta}} - \theta^*)^\top \Sigma_x (\mu_{\hat{\theta}} - \theta^*) + \operatorname{tr}(C_x C_{\hat{\theta}}) + \mu_x^\top C_{\hat{\theta}} \mu_x \xrightarrow[n \to \infty]{} \Delta(\nu_*) + \alpha_*^2 \tag{45}$$

since we know from Corollary 2.2 that $|\mu_x^\top C_{\hat{\theta}} \mu_x| \le O(1/n)$. Then, the identity $\alpha_*^2 = \frac{A(\nu_*)}{\kappa_*^2} \mathbb{E}\big[\xi_y\big(\mu_*^\top x + \alpha_* z, \kappa_*\big)^2\big]$ of Theorem 4.4 combined with (44) yields:

$$\alpha_*^2 = \frac{\nu_*^2\, A(\nu_*)\, \big(\Delta(\nu_*) + \sigma_\varepsilon^2\big)}{1 - \nu_*^2\, A(\nu_*)}.$$

One can then deduce the result of Corollary 5.1 from (45).

### D.2. Reminders of random matrix theory results

We consider the problem of Section 5 in the case $H = \lambda I_p$, which means that $\rho : x \mapsto \lambda \|x\|^2$:

$$\hat{\theta} = \arg\min_{\theta \in \mathbb{R}^p} \frac{1}{n} \|X(\theta - \theta^*) + \varepsilon\|^2 + \lambda \|\theta\|^2, \tag{46}$$

One can easily get with our notation a close formula for our minimizer:

$$\hat{\theta} = \left(\frac{1}{n} X^\top X + \lambda I_p\right)^{-1} \frac{1}{n} X^\top (X\theta^* - \varepsilon) = \frac{1}{n} R(XX^\top \theta^* - X\varepsilon), \tag{47}$$

with the notation

$$R := \left( \frac{1}{n} X X^\top + \lambda I_p \right)^{-1}.$$

Recalling the definition of the resolvent, $Q(\nu) := (\nu C_x + \lambda I_p)^{-1}$, if we consider $\nu_* \in \mathbb{R}$, the unique solution to

$$\frac{1}{\nu} = 1 + \frac{1}{n} \operatorname{tr}(C_x Q(\nu)),$$

a classical result of RMT states that $Q(\nu_*)$ is the deterministic equivalent of $R$. More precisely, given any matrices $A \in \mathbb{R}^{p \times p}$ such that $\|A\|_F \leq O(1)$, one has the following estimates under our concentration assumptions:

$$\forall t > 0 : \mathbb{P}\left( |\operatorname{tr}(A(R - Q(\nu_*)))| \geq t \right) \leq C e^{-cnt^2}, \tag{48}$$

for some constants $C, c > 0$.

Besides, given a deterministic matrix $B \in \mathbb{R}^{p \times p}$ one can also introduce a deterministic equivalent for $RBR$ that we denote:

$$Q_2(B) \equiv QBQ + \frac{\frac{\nu^2}{n} \operatorname{tr}(C_x QBQ)}{1 - \nu^2 A(\nu)} QC_x Q, \tag{49}$$

where we used the notation of introduced in Corollary 5.1:

$$A(\nu) := \frac{1}{n} \operatorname{tr}(QC_x QC_x)$$

This second deterministic equivalent satisfies that if $\|B\| \leq O(1)$ and $\|A\|_F \leq O(1)$:

$$\forall t > 0 : \mathbb{P}\left( |\operatorname{tr}(A(RBR - Q_2(B)))| \geq t \right) \leq C e^{-cnt^2}.$$

### D.3. Estimation of Ridge regression asymptotics through RMT inferences

Since $\varepsilon$ is independent of $X$ with $\mathbb{E}[\varepsilon] = 0$, one can rely on (47) and (48) to get:

$$\mu_{\hat{\theta}} = \mathbb{E}[\hat{\theta}] = \frac{1}{n} \mathbb{E}\left[ RXX^\top \theta^* \right] = \mathbb{E}\left[ (I_p - \lambda R)\theta^* \right]$$
$$= (I_p - \lambda Q)\theta^* + O_{\|\cdot\|}\left( \frac{1}{\sqrt{n}} \right), \tag{50}$$

where we used the identity $R \frac{1}{n} XX^\top = I_p - \lambda R$.

To estimate the covariance of $\hat{\theta}$, let us compute:

$$\mathbb{E}[\theta\theta^\top] = \mathbb{E}\left[ (I_p - \lambda R)\theta^*\theta^{*\top}(I_p - \lambda R) + \frac{1}{n} RX\varepsilon\varepsilon^\top X R \right].$$

Note first from the expression of $Q_2(I_p)$ given in (49) that for any deterministic matrix $S \in \mathbb{R}^{p \times p}$:

$$\mathbb{E}\left[ \frac{1}{n^2} \varepsilon^\top XRSRX\varepsilon \right] = \sigma^2 \mathbb{E}\left[ \frac{1}{n^2} \operatorname{tr}(RXX^\top RS) \right] = \frac{\sigma^2}{n} \mathbb{E}\left[ \operatorname{tr}((I - \lambda R)RS) \right]$$
$$= \frac{\sigma^2}{n} \operatorname{tr}(QS - \lambda Q^2 S) - \frac{\lambda\sigma^2}{n} \frac{\frac{\nu^2}{n} \operatorname{tr}(C_x QSQ)}{1 - \nu^2 A(\nu)} \operatorname{tr}(QC_x Q) + O\left( \frac{1}{\sqrt{n}} \right)$$
$$= \frac{\sigma^2}{n} \operatorname{tr}(QC_x QS)\left( \nu - \frac{\frac{\lambda\nu^2}{n} \operatorname{tr}(QC_x Q)}{1 - \nu^2 A(\nu)} \right) + O\left( \frac{1}{\sqrt{n}} \right)$$
$$= \frac{\sigma^2}{n} \operatorname{tr}(QC_x QS)\left( \frac{\nu - \frac{\nu^2}{n} \operatorname{tr}(QC_x)}{1 - \nu^2 A(\nu)} \right) + O\left( \frac{1}{\sqrt{n}} \right)$$
$$= \frac{\sigma^2 \nu^2}{n} \frac{\operatorname{tr}(QC_x QS)}{1 - \nu^2 A(\nu)} + O\left( \frac{1}{\sqrt{n}} \right),$$

thanks to the identities $\lambda Q = I_p - \nu C_x Q$, $\frac{\lambda}{n}\mathrm{tr}(Q C_x Q) = \frac{1}{n}\mathrm{tr}(C_x Q) + \nu A(\nu)$ and $1 - \frac{\nu}{n}\mathrm{tr}(Q C_x) = \nu$.

Second, note from (49) that:

$$\mathbb{E}[(I_p - \lambda R) S (I_p - \lambda R)]$$
$$= \mathbb{E}[I_p - \lambda R] S \mathbb{E}[I_p - \lambda R] + \frac{\nu^2 \lambda^2}{n} \frac{\mathrm{tr}(C_x Q S Q)}{1 - \nu^2 A(\nu)} Q C_x Q + O_{\|\cdot\|_*}\left(\frac{1}{\sqrt{n}}\right).$$

Then, recalling that $\mu_{\hat\theta} = \mathbb{E}[I_p - \lambda R]\theta^* + O_{\|\cdot\|}\left(\frac{1}{\sqrt{n}}\right)$ one can compute:

$$C_\theta = \frac{\nu^2}{n} \frac{\sigma^2 + \lambda^2 (\theta^*)^\top Q C_x Q \theta^*}{1 - \nu^2 A(\nu)} Q C_x Q + O_{\|\cdot\|_*}\left(\frac{1}{\sqrt{n}}\right)$$

Finally, noting from (50) that $\lambda Q \theta^* = \mu_{\hat\theta} - \theta^* + O_{\|\cdot\|}\left(\frac{1}{\sqrt{n}}\right)$, one can check that:

$$\mathrm{tr}(C_x C_\theta) = \nu^2 A(\nu) \frac{\sigma^2 + (\mu_{\hat\theta} - \theta^*)^\top C_x (\mu_{\hat\theta} - \theta^*)}{1 - \nu^2 A(\nu)} + O_{\|\cdot\|_*}\left(\frac{1}{\sqrt{n}}\right)$$

is exactly the formula of the parameter $\alpha^2$ solution to the minmax problem associated with the convex empirical risk minimization problem.

## E. Proofs of Subsection 6 results

Let us first recall:

**Theorem E.1** (Adamczak (2015, Theorem 2.3)). *Under Assumption 1, for any matrix $A \in \mathbb{R}^{p \times p}$ and all $t \geq 0$:*

$$\mathbb{P}\left( \left| x^\top A x - \mathrm{tr}(\Sigma_x A) \right| \geq t \right) \leq C' e^{-\left( \frac{c' t^2}{\|A\|_F^2} \wedge \frac{c' t}{\|A\|} \right)},$$

*where $C', c' > 0$ do not depend on $n$.*

*Proof of Theorem 6.1 – Gaussian-convoluted expression of $x^\top \hat\theta$.* Denoting $z \sim \mathcal{N}(0,1)$, conditioning on $x$, Assumption 4 provides the concentration in law of $x^\top \hat\theta$ towards $x^\top \mu_{\hat\theta} + \sqrt{x^\top C_{\hat\theta} x}\, z$, which yields:

$$\mathbb{E}\left[ f(x^\top \hat\theta) \right] = \mathbb{E}\left[ \mathbb{E}\left[ f(x^\top \hat\theta) \mid x \right] \right] = \mathbb{E}\left[ \mathbb{E}\left[ f\left( x^\top \mu_{\hat\theta} + \sqrt{x^\top C_{\hat\theta} x}\, z \right) \mid x \right] \right] + O\left(\frac{1}{\sqrt{n}}\right)$$

Now, by Corollary 2.2, we have $\|C_{\hat\theta}\| = O(1/n)$. Consequently, $\|C_{\hat\theta}\|_F \leq \sqrt{p}\|C_{\hat\theta}\| = O(1/\sqrt{n})$. Applying Theorem E.1 with $A = C_{\hat\theta}$ shows that $x^\top C_{\hat\theta} x$ concentrates sharply around its mean:

$$\mathrm{tr}(\Sigma_x C_{\hat\theta}) = \mathrm{tr}(C_x C_{\hat\theta}) + \mu_x^\top C_{\hat\theta} \mu_x.$$

Since $\|\mu_x\| = O(1)$ and $\|C_{\hat\theta}\| = O(1/n)$, the term $\mu_x^\top C_{\hat\theta} \mu_x$ vanishes as $O(1/n)$. Thus, $\mathrm{tr}(\Sigma_x C_{\hat\theta}) = \mathrm{tr}(C_x C_{\hat\theta}) + O(1/n)$, which converges to $\alpha_*^2$ by Theorem 4.3. One finally obtains:

$$\mathbb{E}\left[ f(x^\top \hat\theta) \right] = \mathbb{E}\left[ f\left( x^\top \mu_{\hat\theta} + \alpha_*^2 z \right) \right] + O\left(\frac{1}{\sqrt{n}}\right).$$

$\square$

## F. Proofs of Subsection 7 results

We denote $F_a := F + \mathrm{span}(a)$ and $F_a^\perp := (F + \mathrm{span}(a))^\perp$, and write $P_E$ for the orthogonal projection onto a subspace $E$.

We define $\mathcal{J}_\mu(u) := \mathbb{E}\left[ e_{\mathcal{L}_y}(u^\top x + \alpha z; \kappa) \right] + \rho(u)$, the part minimized by $\mu_*$ in $\mathcal{J}$.

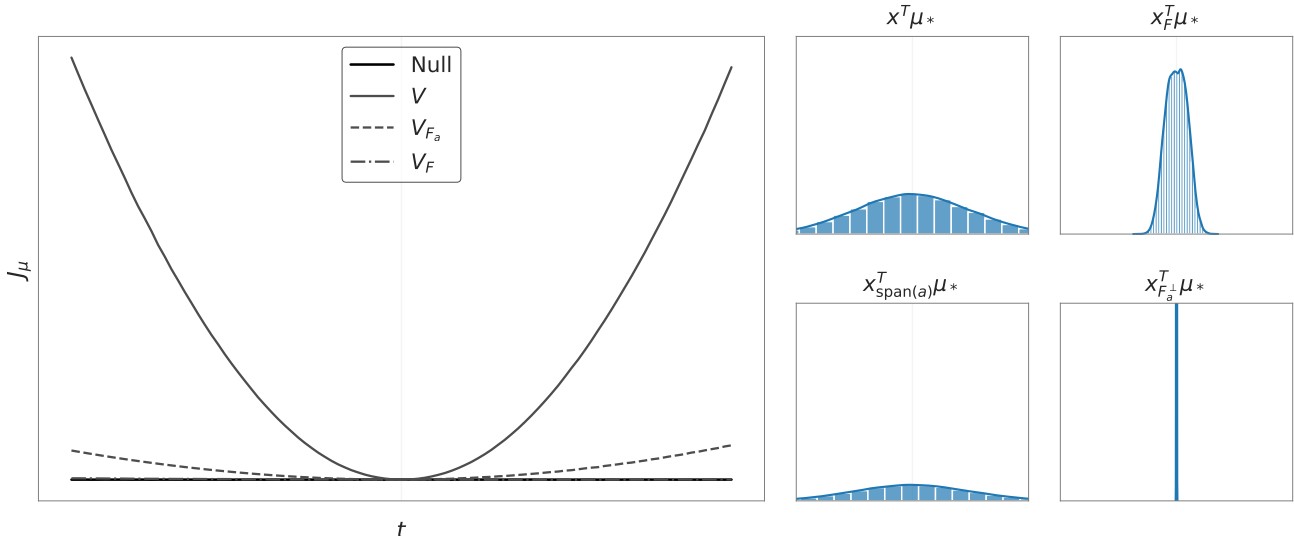

Figure 5. *(left)* Comparison of the values of $t \mapsto \mathcal{J}_\mu(\mu_* + tu)$ for a random direction $u = V$ (solid line), $u = V_{F_a} = P_{F_a}V$ (dashed line), $u = V_F = P_F V$ (dash–dot line), and the null direction $u = 0$ (bold solid line). *(right)* From left to right and from top to bottom we display the following distribution: $x^\top \mu_*$, $(P_F x)^\top \mu_*$, $(P_{F_a} x)^\top \mu_*$ and $(P_{F_a^\perp} x)^\top \mu_*$.

*Proof of Proposition 7.1.* Let the following be $\mu_*$'s decomposition onto $F_a \oplus F_a^{\perp^H}$:

$$\mu_* = \mu_{F_a} + \mu_{F_a^\perp}$$

Where $x \perp^H y \iff x^\top H y = 0$.
Then we have that :

$$
\begin{aligned}
\mathbb{E}[e_l(x^\top \mu_* + \alpha z, \tau, y)] &= \mathbb{E}[\,\mathbb{E}[e_l(x^\top \mu_* + \alpha z, \tau, y)|x_I, y, z]\,] \\
&\geq \mathbb{E}[e_l(\mathbb{E}[x^\top \mu_* + \alpha z \mid x_I, y, z], \tau, y)] \\
&= \mathbb{E}[e_l(x^\top \mu_{F_a} + \alpha z, \tau, y)],
\end{aligned}
$$

where the inequality follows from Jensen's inequality on conditional expectation of convex functions.
We also have that:

$$
\rho(\mu_*) = a^\top \mu_* + \mu_* H \mu_* \tag{51}
$$
$$
= a^\top \mu_{F_a} + \mu_{F_a} H \mu_{F_a} + \mu_{F_a^\perp} H \mu_{F_a^\perp} \tag{52}
$$
$$
\geq a^\top \mu_{F_a} + \mu_{F_a} H \mu_{F_a} \tag{53}
$$

which is always smaller when $\mu_{F_a^\perp} = 0$. Therefore by definition of $\mu_*$ as the minimizer in 4.3, we have:

$$\mu_* \in F_a.$$

$\square$

*Proof of Similarity Results with (Mai & Liao, 2025).* We introduce here (Mai & Liao, 2025)'s setting and results and how we can derive similar results with our equations. First let's define what's a LFMM.

A data point $x$ follows an LFMM if and only if

$$x = \sum_{i=1}^{p} (y s_i + e_i)\, \boldsymbol{v}_i, \tag{54}$$

where $(\boldsymbol{v}_i)_{i=1}^{p}$ are linearly independent vectors, $(s_i)_{i=1}^{p}$ are deterministic coefficients satisfying $s_i = 0$ for $i > q$, with $q$

denoting the number of informative directions, and $(e_i)_{i=1}^p$ are centered unit-variance random variables.
(Mai & Liao, 2025) defines :

$$h_\kappa(t,y) = \frac{1}{\kappa}\big(\text{prox}_{\kappa,l(\cdot,y)}(t) - t\big).$$

They also map the data distribution to the following scalar random variable:

$$u = ym + \tilde{e}\sigma + \sum_{k=1}^q \psi_i e_i,$$

where:

- $\tilde{e} \sim \mathcal{N}(0,1)$ is independent of $y$ and $(e_i)_i$,

- $m, \sigma$, and $(\psi_i)_i$ are deterministic parameters.

We can see further that :

$$\boldsymbol{\mu} = \mathbb{E}[x|y=1] = \sum_{i=1}^q s_i \boldsymbol{v}_i,$$

$$\boldsymbol{\Sigma} = \mathbb{V}[x] = \sum_{i=p}^q \boldsymbol{v}_i \boldsymbol{v}_i^\top.$$

With these definitions, (Mai & Liao, 2025) obtain the following equations:

$$
\begin{cases}
\kappa = \frac{1}{n}\text{tr}(\Sigma Q), \\
\theta = -\mathbb{E}\left[\frac{\partial h_\kappa(u,y)}{\partial u}\right], \\
\gamma = \sqrt{\mathbb{E}[h_\kappa(u,y)^2]}, \\
\eta = \mathbb{E}[y h_\kappa(u,y)], \\
w_i = \mathbb{E}[e_i h(u,\kappa,y)] + \nu v_i^\top Q\boldsymbol{\xi}, \\
\boldsymbol{\xi} = \eta\boldsymbol{\mu} + \sum_{k=1}^q \omega_i \boldsymbol{v}_i, \\
u = y\boldsymbol{\mu}^\top Q\boldsymbol{\xi} + \frac{\gamma}{\sqrt{n}}\sqrt{\text{tr}[(Q\boldsymbol{\Sigma})^2]}\tilde{e} + \sum_{k=1}^q v_i^\top Q\boldsymbol{\xi} e_i.
\end{cases}
\tag{55}
$$

These quantities are then used to define the approximation of $\boldsymbol{\theta}$:

$$\tilde{\boldsymbol{\theta}} = Q\big(\eta\boldsymbol{\mu} + \sum_{k=1}^q w_i \boldsymbol{v}_i + \gamma\boldsymbol{\Sigma}^{\frac{1}{2}}\mathcal{N}(0_p, I_p/n)\big).$$

This implies that

$$\mu_{\boldsymbol{\theta}} = Q\big(\eta\boldsymbol{\mu} + \sum_{k=1}^q w_i \boldsymbol{v}_i\big) = Q\boldsymbol{\xi}$$

$$= Q\big(\eta\boldsymbol{\mu} + \sum_{k=1}^q (\mathbb{E}[e_i h_\kappa(u,y)] + \theta v_i^\top Q\boldsymbol{\xi})\boldsymbol{v}_i\big)$$

$$= Q\big(\eta\boldsymbol{\mu} + \sum_{k=1}^q (\mathbb{E}[e_i h_\kappa(u,y)] + \theta v_i^\top \mu_{\boldsymbol{\theta}})\boldsymbol{v}_i\big).$$

To find their results, we first notice that:

$$\xi_y(t,\kappa) = -\kappa\, h_\kappa(t,y)$$

We start by one of our equations in 4.4 using $h_\kappa(.,.)$ and by abuse of notation we define $u = x^\top \mu_* + \alpha_* \mathcal{N}(0,1)$:

$$\lambda \mu_* = \mathbb{E}[h_\kappa(u,y)x]$$

$$(I - \nu \boldsymbol{\Sigma} Q)Q^{-1}\mu_* = \mathbb{E}[h_\kappa(u,y)x]$$

Given proposition 7.1 and since LFMMs are signal-noise models with $F = \text{Span}\{v_1, \cdots, v_q\}$ then we know that $\mu_* \in \text{Span}\{v_1, \cdots, v_q\}$ which gives:

$$\mu_* = Q \sum_{k=1}^{q} \mathbb{E}[h_\kappa(u,y)(ys_k + e_k)]\boldsymbol{v}_k + \nu Q \boldsymbol{\Sigma} \mu_*$$

$$= Q\eta\boldsymbol{\mu} + Q\sum_{k=1}^{q}\left(\mathbb{E}[h_\kappa(u,y)e_k] + \nu v_k^\top \mu_*\right)\boldsymbol{v}_k.$$

Additionally, we know that :

$$\mathbb{E}\left[h_\kappa(u,y)z\right] = \alpha \mathbb{E}\left[h'_\kappa(\alpha_* z + \mu_* x, y)\right]$$

assuring that $\theta = \nu$.
Finally since:

$$u = x^\top \mu_* + \alpha_* \mathcal{N}(0,1)$$

$$= y\boldsymbol{\mu}^\top \mu_* + \alpha_* \mathcal{N}(0,1) + \sum_{k=1}^{q} v_i^\top \mu_* e_i$$

the expression of $u$ coincides with that of (Mai & Liao, 2025), which concludes the proof of equivalence.  □

# G. Failure of Universality on MNIST Data

This section extends Figure 1 to a structured, real-world dataset and highlights the distinction between two different notions of universality that are often conflated in the literature.

**Two notions of universality.**   The first notion, which we refer to as *score universality*, assumes that for a learned parameter vector $\hat{\theta}$, the decision score $\hat{\theta}^\top x$ is asymptotically Gaussian. This assumption is commonly used to derive performance predictions. The second notion which we refer to as *performance universality* asserts that replacing the data distribution by a Gaussian distribution with matching first- and second-order moments should not affect asymptotic performance. Our results show that both notions can fail.

**MNIST-based data model.**   We consider a binary classification task constructed from MNIST by merging digits $\{3, 6\}$ into class 0 and digits $\{4, 7\}$ into class 1. Samples are represented either by raw pixels, and we compare the original data to a Gaussian surrogate matched in mean and covariance.

**Left panel: failure of score universality.**   The left panel of Figure 2 shows the empirical distribution of the decision score $\hat{\theta}^\top x$. We compare it to a Gaussian distribution with the same empirical mean and variance, as commonly assumed, as well as to our theoretical prediction derived in the non-Gaussian setting. While the Gaussian approximation fails to capture the empirical histogram, our theoretical density closely matches the observed distribution, demonstrating that the score is distinctly non-Gaussian despite moment matching.

**Right panel: consequences for performance.**   The right panel illustrates the impact of this non-Gaussianity on performance predictions. Performance estimates obtained under the Gaussian score assumption (green) exhibit a significant mismatch with empirical results. In contrast, our corrected theoretical predictions accurately track the observed performance. Additionally, we report the performance obtained when the original data are replaced by a Gaussian distribution with matching moments (purple), which also differs substantially from the empirical MNIST performance. This shows that matching first and second moments is insufficient to guarantee universality at the level of generalization performance.

Together, these results demonstrate that both score universality and Gaussian replacement universality can fail in realistic, structured datasets such as MNIST, reinforcing the need for non-Gaussian theoretical corrections.

