# OpenReview forum: "Characterization of Gaussian Universality Breakdown in High-Dimensional Empirical Risk Minimization"
_ICML.cc/2026/Conference — ICML 2026 regular_

### Official Review · Reviewer_zGDu · 2026-03-11

**Soundness:** 3
**Presentation:** 2
**Significance:** 2
**Originality:** 2
**Overall Recommendation:** 4
**Confidence:** 3

**Summary:**

This paper studies the asymptotic behavior of high-dimensional convex empirical risk minimization (ERM) under potentially non-Gaussian data designs satisfying a concentration assumption. Their goal is to characterize the distribution of the test score $x^\top \hat{\theta}$ and the resulting prediction performance in the proportional high-dimensional regime. The work focuses on convex losses and smooth regularizers, and develops a framework extending the CGMT approach beyond the standard Gaussian design setting.

The key idea is to derive deterministic equivalents for the mean and covariance  of the ERM minimizer  $\hat{\theta}$, and use them to characterize the asymptotic distribution of the prediction score. The authors show that under concentration assumptions on the data and smoothness conditions on the loss and regularizer, the test score behaves asymptotically as $x^\top \hat{\theta} \approx x^\top \mu_* + \alpha_* z$ where $z\sim\mathcal N(0,1)$ and $(\mu_*,\alpha_*)$ are solutions of deterministic and explicit saddle-point equations derived from the ERM objective. More precisely, the main results are:

- Quadratic universality of smooth regularizers: Any $C^2$ regularizer is asymptotically equivalent, for the statistical behavior of $\hat{\theta}$, to a quadratic surrogate determined by its Hessian at the origin and its gradient at the mean.
- Deterministic characterization via a min–max problem: The asymptotic statistics of $\hat{\theta}$ are obtained from the solution of deterministic and explicit self-consistent equations which depends on the data distribution only through expectations involving the scalar variable $\mu^\top x + \alpha z$ .
- Performance universality in special cases: For ridge regression under a linear model, the asymptotic generalization error depends only on the first two moments of the features, implying performance universality even when the score distribution itself is non-Gaussian.

Moreover, the authors discuss conditions for score universality and a geometrical characterization for when Gaussian score universality is expected to hold. Overall, the paper provides a unified framework for analyzing high-dimensional ERM beyond the Gaussian design assumption. It clarifies the relationship between different notions of universality (minimizer, score, and performance) and identifies precise conditions under which Gaussian approximations for the prediction score are valid or break down.

**Compliance With Llm Reviewing Policy:**

Affirmed.

**Final Justification:**

My initial assessment highlighted a clear paper addressing an interesting question in exact asymptotics, with potentially interesting contributions towards unifying different notions of universality for ERM.

However, I had concerns regarding (i) the lack of explicit discussion of the scope and limitations of the assumptions, (ii) the need for a more concrete interpretation of the main results and their novelty relative to existing Gaussian and non-Gaussian settings, and (iii) a somewhat superficial positioning with respect to closely related literature. The authors rebuttal addressed these points satisfactorily.

While some of these improvements remain to be fully reflected in a revised version of the manuscript, they provide a clear list of points for revision, which is convincing. Overall, the rebuttal has positively updated my evaluation, leading me to revise my recommendation from "weak reject" to "weak accept|.

**Key Questions For Authors:**

Please refer to the questions in *Weaknessess*.

**Limitations:**

Limitations are only partially discussed in the paper. Please refer to the first point in *Weaknessess*.

**Strengths And Weaknesses:**

**Strengths**: Overall, the paper is clearly structured and the narrative is clear. The main results are also clearly highlighted. Universality in an important topic in the context of exact asymptotics, since provides stronger grounding for formulas which are typically derived under unrealistic data assumptions.  Several papers have investigated this question over the past ten years, many of which at ICML. Therefore, it is a timely and relevant topic which is of interest to the exact asymptotics community at ICML. This paper provides some potentially interesting contributions in this directions, particularly in the relationship between different notions of universality appearing in the literature.

**Weaknessess**: In my reading, the three main weaknesses of this work are:
- Lack of discussion on the limitations. The paper makes a few broad assumptions, and from the reading it is unclear what are their scope. For example:
    - In Assumption 1, the authors give an example where it holds. What about cases which it fails?
    - Why is Claim 4.1 not a theorem? What are the challenges in proving it?
    - The text suggests that Theorem 4.3 follows from Claim 4.1. Do you need to assume Claim 4.1 to hold in Theorem 4.3 or it is a particular case under which you can make it rigorous?
    - The author says that assumption 3 does not hold for LASSO, but it holds for ridge. Are there other cases in which it holds? E.g. $\ell_{p}$ with $p>1$? Anisotropic regularization $||\theta||_{\Lambda}$ with $\Lambda$ satisfying some properties? From the final formulas, it seems that some delocalization is important for the universality result on the regularizer to hold, since it only depends on traces.
    - The authors say they are not aware of any example where Assumption 4 does not hold. Do the authors believe this can be proven? What are the challenges in proving this?

- Lack of a concrete discussion of the main results. The paper is technically dense, and it would be useful to have some concrete discussion to stress the novelty of the results. For instance:
    - Are there limits in which their formula reduce to the asymptotic Gaussian formula?
    - The authors put forward that their result hold for non-Gaussian covariates and precisely characterize the break of universality. Although less common, non-Gaussian covariates were studied in different contexts already in the proportional regime in the literaure, e.g. single and multi-layer random features, Gaussian mixture, one-step of SGD, etc. For some of these examples, universality holds and for some it breaks. Do you have concrete examples covered by your results which are not discussed in the literature?

- Superficial discussion of the related literature. Although the authors make an effort to cite the universality literature (which I acknowledge is vast), the comparison with some of the closer works remain vague, which does not help understanding the novelty of the results. Also, some important references, both old and new, are missing. More precisely:
    - **Major**:
    - What the authors call "score universality"/1d CLT and its consequence for risk universality was first discussed in (Goldt et al., 2022), concurrently to (Hu & Lu 2022). In particular, both appeared before (Montanari & Saeed 2022), this should be more throughoutly acknowledged when referring to this connection.
    - Discussion in L065-L072: The most general CGMT proof for convex ERM with Gaussian covariates appeared in (Loureiro et al., 2021b). This work also discussed empirically the limitations of these formulas, even in the case of unimodal distributions (e.g. see Fig. 3 therein)
    - Different works have studied how to derive exact asymptotic results for problems in which Gaussian universality does not work, e.g. mixture distributions (Dandi et al., 2023), random features + one-step of SGD (Cui et al., 2024; Dandi et al., 2024), generalized linear models with quadratic features (Wen et al., 2025). A common denominator in these works is the so-called Conditional Gaussian Equivalence (cGET). This holds for problems in which the projection of the non-Gaussian component in the task on the data is low-dimensional, with the remainder of the randmness being enough to ensure concentratation in the limit. A deeper discussion on the relationship between cGET and the results in this work is pertinent, in particular with respect to Section 6, which seems to state a very similar condition.
    - **Minor**:
    - The discussion of Gaussian universality in the context of GMMs first appeared in (Gerace et al., 2022), before (Pesce et al., 2023).
    - Discussion in L336-337: High-dimensional asymptotics for mixture models was also discussed in (Mignacco et al., 2020, Wang & Thrampoulidis 2020; Loureiro et al., 2021).

- (Goldt et al., 2022) The Gaussian equivalence of generative models for learning with shallow neural networks, https://proceedings.mlr.press/v145/goldt22a.html
- (Loureiro et al., 2021b) Learning curves of generic features maps for realistic datasets with a teacher-student model, https://proceedings.neurips.cc/paper/2021/hash/9704a4fc48ae88598dcbdcdf57f3fdef-Abstract.html
- (Gerace et al., 2022) Gaussian universality of perceptrons with random labels, https://arxiv.org/abs/2205.13303
- (Mignacco et al., 2020) The Role of Regularization in Classification of High-dimensional Noisy Gaussian Mixture, https://proceedings.mlr.press/v119/mignacco20a.html
- (Wang & Thrampoulidis 2020) Binary Classification of Gaussian Mixtures: Abundance of Support Vectors, Benign Overfitting and Regularization, https://arxiv.org/abs/2011.09148
- (Cui et al., 2024) Asymptotics of feature learning in two-layer networks after one gradient-step, https://arxiv.org/abs/2402.04980
- (Dandi et al., 2024) A Random Matrix Theory Perspective on the Spectrum of Learned Features and Asymptotic Generalization Capabilities, https://arxiv.org/abs/2410.18938
- (Wen et al., 2025) When does Gaussian equivalence fail and how to fix it: Non-universal behavior of random features with quadratic scaling, https://arxiv.org/abs/2512.03325

---

> ### Author Rebuttal · Authors · 2026-03-27
>
> Thank you for the detailed review and for the very useful reference list.
>
> To make navigation easy, we use the same shared-response labels as in the other rebuttals.
>
> 1. **Limitations and scope**
>
>    We agree that the scope should be stated more clearly. Assumption 1 is a concentration assumption: it covers, for example, covariates of the form $x=\Phi(z)$ with Gaussian latent $z$ and Lipschitz $\Phi$, as well as class-conditional Lipschitz maps in classification, but it does not cover heavy-tailed or non-concentrated designs. We will add this explicitly.
>
>    On Claim 4.1 and Assumptions 3/4, please see **Shared response C**, **Shared response B**, and **Shared response D**. Since you ask specifically about Assumption 3, let us also stress here that anisotropic quadratic regularization,
>    $$
>    \rho(\theta)=\frac12\theta^\top H\theta,
>    $$
>    is fully covered (not only isotropic ridge), since $\nabla^3\rho\equiv 0$. More generally, Assumption 3 allows smooth non-separable perturbations whose third-order interactions are asymptotically negligible in aggregate.
>
> 2. **Concrete discussion of the main results**
>
>    Please see **Shared response A** in the rebuttal to Reviewer zqV1. We agree that the current draft should better explain the broader goal of the paper.
>
>    One central purpose of our work is **unification**: rather than analyzing one specific feature model, we identify ERM-level assumptions under which one and the same deterministic min-max / fixed-point description governs a broad class of smooth ERM problems, across many data models and losses, while also revealing when Gaussian universality fails.
>
>    In the Gaussian case, or more generally whenever $x^\top\mu_*$ is Gaussian, our score description reduces to the usual Gaussian fixed-point picture.
>
>    Beyond classical Gaussian design, Assumption 1 covers concentrated non-Gaussian representations such as Lipschitz feature maps of Gaussian latents. Our bimodal synthetic example and the MNIST experiment are concrete cases where the same $(\mu_*,\alpha_*)$ description remains predictive while the score itself is non-Gaussian.
>
>    More importantly, it also covers observed covariates of the form
>    $$
>    x=\Phi_y(z),\qquad z\sim\mathcal N(0,I_d),\qquad y\in\{\pm1\},
>    $$
>    with $\Phi_y$ bounded-Lipschitz, for example
>    $$
>    \Phi_y(z)=c_1\tanh(c_2 z+b_y),
>    $$
>    for some constants $c_1,c_2\in \mathbb R$. This gives a class of concentrated, class-conditional nonlinear latent models whose informative projection can remain genuinely non-Gaussian. To our knowledge, the cited exact-asymptotic ERM literature does not provide a direct characterization for this class at the level of the observed covariates without first invoking a Gaussian or conditional-Gaussian surrogate under a one-dimensional CLT or an approximate-Gaussian-projection hypothesis. This is one concrete sense in which our framework is more general.
>
>    The anonymous repository reproduces these experiments and includes additional notebooks; we will point readers to it more clearly.
>
> 3. **Related work / chronology / cGET**
>
>    We agree that the related-work section should be strengthened. We will explicitly acknowledge Goldt et al. (2022) together with Hu & Lu (2022) when discussing score universality / one-dimensional CLTs and their implications for performance universality, and clarify the chronology relative to Montanari & Saeed (2022). We will also expand the discussion around Loureiro et al. (2021b), Gerace et al. (2022), mixture-model work, and the more recent random-feature / cGET-related papers you listed.
>
>    We also agree that the connection with cGET should be made explicit. In fact, the connection is direct at the **score level**: our Section 6 result
>    $$
>    x^\top\hat\theta \approx x^\top\mu_*+\alpha_* z
>    $$
>    has exactly the cGET structure, where the only potentially non-Gaussian part is the informative projection $x^\top\mu_*$, while the remaining fluctuations collapse to a Gaussian scalar. Under the subspace confinement result of Section 7, this becomes even closer in spirit to cGET, since only the projection onto the informative subspace $F+\mathrm{span}(a)$ can retain non-Gaussianity.
>
>    The difference is mainly one of formulation: cGET is usually introduced through an explicit hybrid surrogate for a specific model class, whereas our goal is to identify ERM-level assumptions under which the same deterministic characterization applies across a broad class of smooth ERM problems. We will make both the overlap and the difference explicit in the revision.
>
>    Wen et al. is also a very recent preprint, which is one reason it was missing from our initial draft; we agree that it should be added.
>
> 4. We appreciate the references and will incorporate them.

---

> > ### Author Rebuttal · Reviewer_zGDu · 2026-04-03
> >
> > I thank the authors for their rebuttal. My concerns have been addressed, and I am happy with the list of changes proposed. I think they will clarify some of the points and strength the manuscript. I will be updating my score to 4.

---

### Official Review · Reviewer_mdB9 · 2026-03-12

**Soundness:** 3
**Presentation:** 3
**Significance:** 2
**Originality:** 3
**Overall Recommendation:** 4
**Confidence:** 2

**Summary:**

This paper studies high-dimensional empirical risk minimization (ERM) under non-Gaussian design and examines the limits of Gaussian universality, the practice of approximating non-Gaussian quantities (e.g., covariates, scores, or estimators) with the appropriate Gaussian ones. The authors analyze when such approximations hold and fail for convex ERM problems. Namely, building on ideas related to the Convex Gaussian Min–Max Theorem (CGMT), they propose a framework for non-Gaussian settings, under certain concentration assumptions, that yields a min–max characterization and fixed-point equations for approximating the mean and covariance of the ERM estimator, thereby identifying conditions under which Gaussian universality holds or breaks down. They further show that, under smoothness assumptions, a general $C^2$ regularizer can be asymptotically replaced by a quadratic surrogate defined by its gradient and Hessian, which simplifies the analysis. Numerical simulations are done to illustrate the theoretical predictions.

**Compliance With Llm Reviewing Policy:**

Affirmed.

**Final Justification:**

My main concerns relate to the level of theoretical rigor and clarity. The central technical component (Claim 4.1) remains unproven and is presented as a conjectural extension of CGMT. While this weakens the formal guarantees, the rebuttal clearly clarified that subsequent results are conditional on this step and improved the transparency of the contribution. Similarly, assumptions such as minimizer universality are not fully justified, but the authors provided useful intuition and outlined plausible routes toward formalization. Importantly, the rebuttal addressed several of my initial concerns in a constructive way.

Therefore, I support a weak accept, as I believe the paper offers interesting ideas and a unifying perspective that are likely to be valuable to the community, even if some components remain to be fully formalized.

**Key Questions For Authors:**

**Questions**
1. My main question is can the authors identify specific technical reasons preventing a proof of Claim 4.1? Is it a conjectured extension, something provable under extra structural assumptions, or a reformulation of an existing result? Even partial results here would significantly strengthen the paper.
2. Can the authors identify concrete model classes where Assumption 4 can be proved?
3. Which parts of the analysis break down when the regularizer is non-smooth, e.g., lasso? Is the quadratic surrogate argument the main obstacle, or are there deeper issues?

**Typos encountered**

* Page 1: seminar work -> seminal work
* Page 2: relies some form ... -> relies on some form ...
* Page 8: he underlying heuristic - > the underlying heuristic
* Page 11: Proof of Subsection 4 results -> Proof of Subsection 3 results
* Page 14: Proof of Subsection 3 results -> Proof of Subsection 4 results

**Limitations:**

Yes

**Strengths And Weaknesses:**

**Strengths**

1. The paper is clearly written.

2. The problem is well-motivated, i.e. the question of when Gaussian universaility holds or breaks down is very important, and one of the results is veary elegant the test score is Gaussian if and only if the projection of covariates onto the estimator mean is Gaussian.

3. The paper seems well situated within the existing literature and both extends and recovers several known results.

4. The quadratic universality result is conceptually interesting and may have useful practical implications.

**Weaknesses**

1. The technical contribution is not completely clear. Namely, the central CGMT extension, Claim 4.1, is not formally proven. Appendix C derives consequences but does not provide a rigorous proof of the claim itself. Since Theorems 4.3 and 4.4 rely on this claim, the lack of a proof or further comments on settings when it would hold weaken the theoretical contribution.

2. The score characterization in Section 6 relies on Assumption 4 regarding minimizer universality, but there is lack of rigorous justification. As a result, one of the paper’s main results, Theorem 6.1 remains conditional. It would strengthen the paper to clarify when minimizer universality is expected to hold or fail, and prove/disprove in certain settings.

3. The scope is also narrower than the title and abstract suggest. The assumptions require smoothness and strong convexity of the regularizer, with additional C^3-type control for the quadratic approximation argument. This excludes many practically important ERMs, especially non-smooth penalties such as lasso. So while the paper is about “general non-Gaussian designs,” it is not yet a broadly general ERM theory.

---

> ### Author Rebuttal · Authors · 2026-03-27
>
> Thank you for pinpointing the main technical issues. We answer in the order of your questions and reuse the same labels as in the other rebuttals.
>
> 1. **Claim 4.1**
>
>    Please see **Shared response C** in the rebuttals to Reviewers w5Yu. We agree that the paper should make this much more explicit: Claim 4.1 is the only non-rigorous comparison step, and Theorems 4.3 and 4.4 are downstream consequences of it. We will revise the wording accordingly.
>
> 2. **Assumption 4**
>
>    **Shared response D: Assumption 4 / minimizer universality**
>
>    The role of Assumption 4 is limited: it is used only in Section 6 to turn the mean/covariance characterization into a full score law; the non-Gaussian min-max / fixed-point characterization in Section 4 does not rely on it.
>
>    The heuristic comes from the linearized KKT relation
>
>    $$
>    H_\rho(\hat\theta - \mu_{\hat\theta}) \approx -\nabla\rho(\mu_{\hat\theta})   - \frac{1}{n}\sum_{i=1}^n \mathcal L'_{y_i}(x_i^\top \hat\theta)x_i.
>    $$
>
>    This suggests that $\hat\theta - \mu_{\hat\theta}$ behaves like an average of many weakly dependent terms. The difficulty is that the summands depend on $\hat\theta$ itself, so a standard CLT does not apply directly (even in the case of a linear model and $\ell_2$ losses). One needs to control both the fluctuations of the optimizer and the dependence of the summands on that same optimizer. As explained in the text, we believe it can be shown with a Lindeberg-type argument relying on a leave-one-out expression of $\hat \theta$ that allows to characterize fully the dependence of $\hat \theta$ on each data $x_i$. Although we still haven't fully proven it, it should be accessible for all concentrated model and smooth losses satisfying Assumption 1,2,3 of our paper.
>
>    Our point here is to isolate this remaining Gaussian universality property as the extra step needed for a full score law. We will revise the text to make this conditional status much more explicit.
>
> 3. **Nonsmooth penalties / scope**
>
>    **Shared response E: nonsmooth penalties and scope**
>
>    The limitation to smooth losses and smooth regularizers is broader than just the quadratic surrogate. Smoothness and strong convexity enter in three places:
>
>    1. concentration and uniqueness of $\hat\theta$;
>    2. the quadratic surrogate itself, via a second/third-order Taylor expansion;
>    3. the linearized fluctuation heuristic behind Assumption 4.
>
>    Empirically, in our experiments (see the elastic-loss notebook in the anonymous repository), the predictions remain accurate for non-smooth loss functions such as  $\ell_1$ loss, but we observe qualitatively different behaviors when the regularizer itself is non-smooth (e.g. $\ell_1$ regularization). This is consistent with the fact that our approach relies on a quadratic (Taylor) approximation of the regularizer, which is surely not valid in the non-smooth case.
>    These observations suggest that extending the theory to non-smooth regularizers would likely require a different route, closer to direct nonsmooth CGMT arguments.
>
>    We will make this scope much more explicit in the limitations paragraph and in the abstract/introduction: concentrated designs, smooth losses, and smooth strongly convex regularizers.
>
> 4. Thank you for the typo list. We will correct all of them.

---

> > ### Author Rebuttal · Reviewer_mdB9 · 2026-04-03
> >
> > Thank you to the authors for the rebuttal and clarifications. I believe my concerns have been addressed and that proposed revisions will improve the paper’s accessibility. I increase my score to 4.

---

### Official Review · Reviewer_w5Yu · 2026-03-13

**Soundness:** 3
**Presentation:** 4
**Significance:** 3
**Originality:** 3
**Overall Recommendation:** 5
**Confidence:** 3

**Summary:**

This paper investigates high-dimensional convex empirical risk minimization to formalize when the widely used Gaussian universality assumption fails on non-Gaussian data. The authors heuristically extend the Convex Gaussian Min-Max Theorem through a quadratic surrogate for smooth regularizers, yielding deterministic equations for the estimator's asymptotic performance. Theoretically, they establish that a test score's Gaussianity hinges entirely on the projection of a test covariate onto an asymptotic signal proxy vector. Finally, experiments on synthetic and MNIST datasets validate this framework.

**Compliance With Llm Reviewing Policy:**

Affirmed.

**Final Justification:**

The authors' rebuttal addressed my questions. I maintain my initial assessment.

**Key Questions For Authors:**

1. What are the precise conditions under which claim 4.1 could be made rigorous, or what are the fundamental obstacles?
2. The author mentions that Assumption 3 is weaker that the standard weak separability assumption. Could you expand this discussion, providing examples on non-separable regularizers satisfying the assumption?
3. Could you provide more general conditions allowing performance universality to survive the breakdown of score universality?

**Limitations:**

Yes

**Strengths And Weaknesses:**

### Strengths

The authors investigate the ambitious goal of characterizing the breakdown of Gaussian universality, which is a fundamental question for the interpretation of asymptotics derived via AMP, RMT, and CGMT. The presentation is well-structured, with clear definitions of the three universality notions and careful positioning within the existing literature. The paper is overall technically sound and the results are interpretable. Theorems 3.3 and 7.1 add geometric intuition that simplifies the verification of universality in structured models. The empirical validation further corroborates the paper's claim.

### Weaknesses
I have not identified any major weaknesses.
1. Part of the results rely on the strong assumption of minimizer universality, which restricts the analysis to regimes where the estimators is sufficiently well-behaved. However, this limitation is transparently acknowledged and justified by the lack of known counterexamples.
2. The font size in the figures' labels and legends is very small and hardly readable.

---

> ### Author Rebuttal · Authors · 2026-03-27
>
> Thank you for the careful and positive reading. We will enlarge all figure labels and legends.
>
> To make navigation easy, we use shared-response labels that are referenced in other rebuttals.
>
> 1. **Claim 4.1 and the obstacles to a fully rigorous statement**
>
>    **Shared response C: Claim 4.1 and Theorems 4.3--4.4**
>
>    We agree that the draft should separate more clearly what is rigorous from what is conjectural. Claim 4.1 is meant to isolate the non-Gaussian extension of the CGMT comparison step; once that comparison principle is granted, the downstream convex analysis leading to Theorems 4.3 and 4.4 is explicit.
>
>    The main obstacle is that standard CGMT relies on exact Gaussian rotational invariance to replace the bilinear form $\theta^\top A w$ by an auxiliary Gaussian process. For concentrated non-Gaussian columns, one needs a universality theorem for both the value and the optimizer of the saddle problem, together with control of the data-dependent shift $\mu_{\hat\theta}$. This is a genuinely difficult theoretical problem: one can check indeed that the original CGMT theorem by Gordon requires already delicate probabilistic arguments. At present, we do not have a rigorous proof of this extension, so we state it as a conjectural step. We still believe this formulation is useful because it isolates the exact missing ingredient and leads to concrete predictions that are consistently supported by our experiments.
>
>    We will revise the wording so that Theorems 4.3 and 4.4 are explicitly presented as consequences of Claim 4.1, making the location of the heuristic step fully transparent.
>
> 2. **Assumption 3 and non-separable examples**
>
>    Please see **Shared response B** in the rebuttals to Reviewers zqV1. In particular, we will add explicit non-separable examples such as
>    $$
>    \rho(\theta) = \frac12\theta^\top H \theta + \phi\left(p^{-1/2} u^\top \theta\right),
>    $$
>    and clarify why Assumption 3 is weaker than the usual weak-separability condition.
>
> 3. **When performance universality can survive score breakdown**
>
>    A sufficient mechanism is that the target metric depends on the score only through low-order moments determined by $(\mu_*, \alpha_*)$, rather than through the full score law. The ridge / squared-loss linear-model corollary is the clean example of this phenomenon (and might be the only one); please see **Shared response A** in the rebuttals to Reviewers zqV1.
>
>    As a concrete empirical illustration, the anonymous repository includes the [Notebook 3](https://anonymous.4open.science/r/Empirical-risk-minimization-asymptotics-9162/Notebook3_universality_through_elastic_loss.ipynb) sweeping the elastic loss
>    $$
>    \mathcal L_\eta = (1-\eta)\ell_2 + \eta \ell_1
>    $$
>    on a bimodal model. At $\eta = 0$, squared loss shows performance universality even though the score is non-Gaussian; as $\eta$ increases toward the non-smooth $\ell_1$ regime, the Gaussian risk proxy degrades. See the figure at the bottom of the notebook. We present this notebook only as an illustration of the boundary of the theory, not as a theorem beyond the smooth setting.

---

> > ### Author Rebuttal · Reviewer_w5Yu · 2026-04-03
> >
> > Thank you for your rebuttal and clarification, all my questions have been answered.

---

### Official Review · Reviewer_zqV1 · 2026-03-18

**Soundness:** 3
**Presentation:** 1
**Significance:** 4
**Originality:** 4
**Overall Recommendation:** 4
**Confidence:** 1

**Summary:**

This paper studies the extension of the Convex Gaussian min-max theorem to non-Gaussian settings. The claim is that this ascertains the scope of Gaussian universality for ERM’s. Gaussian universality essentially means that learning problems under the ERM paradigm can essentially be analyzed as if the data is from a Gaussian distribution with same parameters as the original distribution.

**Compliance With Llm Reviewing Policy:**

Affirmed.

**Final Justification:**

Conditioned on the authors improving the exposition, I have updated my score. I think it is a notable contribution and have upgraded my score to a weak accept.

**Key Questions For Authors:**

1. Can you please outline a concrete practical use case and implications of the current work on it?
2. I could get the overall gist of the paper and proofs. But could you explain, in simple terms, the content of Theorem 4.3. As stated in the paper, it is a bit dense.
3. Assumption 3 seems to be artificial. Can you please say a bit more as to why it makes sense and is natural?

**Limitations:**

Yes

**Strengths And Weaknesses:**

Strengths
1. The work is a contribution to the decade long line of work regarding the Gaussian universality for ERM’s and makes several non-trivial strides in the state of the art knowledge in the area.

Weaknesses -

1. The presentation is extremely dense and almost unverifiable.
2. A major weakness is that even the problem statement is not clear to a computer science researcher outside of the field despite significant effort. Pointers to surveys, avoiding usage of technical terms without precedence especially in the introduction and "our contributions" section can help with the presentation.
3. Nit: In page 2, it should be ‘ERM’ solution instead of ‘EMR’.

---

> ### Author Rebuttal · Authors · 2026-03-27
>
> Thank you for the positive assessment of the significance and originality. We agree that the current exposition is too dense for a broader ML audience. In the revision, we will simplify the introduction, add a short roadmap with the three notions of universality stated upfront, include broader pointers to CGMT/AMP/RMT, and add a plain-language explanation of Theorem 4.3.
>
> We also understand the concern about verifiability. The anonymous GitHub repository linked in the paper was meant to make the claims easier to check: it reproduces the synthetic and MNIST experiments and includes additional notebooks for several models and losses. We will point readers to it more clearly in the paper as empirical support for the theory.
>
> To make navigation easy, we answer in the order of your questions and use the same labels for answers that also appear in other rebuttals.
>
> 1. **Practical use case and simple meaning of Theorem 4.3**
>
>    **Shared response A: practical meaning / Theorem 4.3 / performance universality**
>
>    A concrete use case is high-dimensional classification or regression on structured non-Gaussian features, for example mixture models, random-feature-type maps, or real embeddings such as MNIST features, where Gaussian asymptotics are often used to tune regularization or predict test error. Our result tells us when that shortcut is reliable and when it is not.
>
>    In simple terms, Theorem 4.3 says that the random ERM can be summarized by two deterministic quantities: a mean proxy $\mu_{\ast}$, asymptotically equal to $\mu_{\hat\theta} = \mathbb E[\hat\theta]$, and a fluctuation size $\alpha_\ast$, asymptotically given by $\alpha_*^2 \approx \mathrm{tr}(C_x C_{\hat\theta})$. These two quantities are obtained from an explicit min-max problem, or equivalently from the fixed-point system in Theorem 4.4.
>
>    Under Assumption 4, Section 6 then shows that the test score behaves like
>    $$
>    x^\top \hat\theta  \approx  x^\top \mu_\ast + \alpha_\ast z,
>    $$
>    with $z \sim \mathcal N(0,1)$ independent of $(x,y)$. Hence score Gaussianity holds if and only if the informative projection $x^\top \mu_\ast$ is Gaussian.
>
>    This matters in practice because a Gaussian proxy can miss skewness or bimodality of the score and therefore misestimate classification error. Our synthetic mixture and MNIST experiments illustrate this. At the same time, performance universality can still survive score non-universality when the metric depends only on low-order score moments rather than on the full score law; the ridge / squared-loss linear-model corollary is the clean example of this mechanism.
>
> 2. **Assumption 3**
>
>    **Shared response B: Assumption 3**
>
>    Assumption 3 does not require separability. It is a regularity condition on the third derivatives of $\rho$: it says that cross-coordinate third-order interactions are small enough at the $p=\Theta(n)$ fluctuation scale relevant to $\hat\theta - \mu_{\hat\theta}$. This is exactly the scale at which the quadratic surrogate is used, and it explains why only trace-level second-order quantities remain in the final formulas.
>
>    The assumption is automatic for all quadratic regularizers, including anisotropic ridge ($\Vert\theta\Vert_\Lambda$), since $\nabla^3 \rho \equiv 0$, and for separable smooth penalties (like the $\ell_p$ regularizations), since the off-diagonal part of the third-derivative tensor is zero. It also allows genuinely non-separable examples such as
>    $$
>    \rho(\theta) = \frac12\theta^\top H \theta + \phi\left(p^{-1/2} u^\top \theta\right),
>    $$
>    with $H \succeq \kappa I$, $\phi \in C^3$ with bounded third derivative, and $\|u\| = O(1)$. Here the extra third-order coupling is low-rank and satisfies the assumption. This class combines a strongly convex quadratic baseline with a nonlinear penalty acting only along one informative direction, hence it is genuinely non-separable while keeping third-order interactions low-rank. We concede, however, that we are unaware of any prior appearance in the literature of the term $\phi\left(p^{-1/2} u^\top \theta\right)$  within the proposed regularizations.
>
>    We will add such examples and clarify more explicitly why this is weaker than the standard weak-separability assumption: we control only the aggregate off-diagonal third-order effect, not a coordinate-wise decomposition.
>
> 3. We will also fix the typo `EMR` $\to$ `ERM`.

---

> > ### Author Rebuttal · Reviewer_zqV1 · 2026-04-03
> >
> > Thank you for the rebuttal.

---

### Decision · Program_Chairs · 2026-04-30

**Decision:**

Accept (regular)

**Comment:**

This paper studies an important and timely question: when Gaussian universality holds or breaks down for high-dimensional convex ERM under non-Gaussian designs. Reviewers agreed that the problem is relevant and that the paper offers an interesting unifying perspective on minimizer, score, and performance universality.

The main concern was the level of rigor of the central non-Gaussian CGMT extension (Claim 4.1), together with the conditional nature of some downstream results and the need for clearer discussion of scope, assumptions, and related work. After rebuttal, however, all reviewers indicated that their concerns were adequately addressed, and the final assessments converged to 4/4/4/5.

My recommendation is thus to accept the paper. The work appears technically meaningful and likely to be useful to the exact-asymptotics community, even though some central steps remain conjectural rather than fully proved.

It is important, however, that the revision should make this conditional status fully explicit, sharpen the discussion of assumptions and limitations, and strengthen the exposition and positioning relative to prior work.